# Regulation of *TRI5* expression and deoxynivalenol biosynthesis by a long non-coding RNA in *Fusarium graminearum*

Panpan Huang[1,2], Xiao Yu[1], Huiquan Liu ®[1], Mingyu Ding[1], Zeyi Wang[2], Jin-Rong Xu ®[2] ✉ & Cong Jiang ®[1] ✉

Deoxynivalenol (DON) is the most frequently detected mycotoxin in cereal grains and processed food or feed. Two transcription factors, Tri6 and Tri10, are essential for DON biosynthesis in *Fusarium graminearum*. In this study we conduct stranded RNA-seq analysis with *tri6* and *tri10* mutants and show that Tri10 acts as a master regulator controlling the expression of sense and antisense transcripts of *TRI6* and over 450 genes with diverse functions. *TRI6* is more specific for regulating *TRI* genes although it negatively regulates *TRI10*. Two other *TRI* genes, including *TRI5* that encodes a key enzyme for DON biosynthesis, also have antisense transcripts. Both Tri6 and Tri10 are essential for *TRI5* expression and for suppression of antisense-*TRI5*. Furthermore, we identify a long non-coding RNA (named RNA5P) that is transcribed from the *TRI5* promoter region and is also regulated by Tri6 and Tri10. Deletion of RNA5P by replacing the promoter region of *TRI5* with that of *TRI12* increases *TRI5* expression and DON biosynthesis, indicating that RNA5P suppresses *TRI5* expression. However, ectopic constitutive overexpression of RNA5P has no effect on DON biosynthesis and *TRI5* expression. Nevertheless, elevated expression of RNA5P in situ reduces *TRI5* expression and DON production. Our results indicate that *TRI10* and *TRI6* regulate each other's expression, and both are important for suppressing the expression of RNA5P, a long non-coding RNA with *cis*-acting inhibitory effects on *TRI5* expression and DON biosynthesis in *F. graminearum*.

Fusarium head blight (FHB) is a devastating disease of wheat and barley that threatens global food security. *Fusarium graminearum* is one major causal agent of FHB and it also infects corn, millets, and other grain crops[1–4]. In addition to causing significant yield losses, *F. graminearum* is a producer of deoxynivalenol (DON), which is the most frequently detected mycotoxin in cereal grains and food or feed processed from cereal grains[5]. As a type B trichothecene mycotoxin, DON is inhibitory to eukaryotic protein synthesis and harmful to human or animal health. It is also phytotoxic and DON production plays a critical role during plant infection[6,7]. The gene that encodes the Tri5 trichodiene synthase is the first virulence factor functionally characterized by molecular genetic studies in *F. graminearum*[8]. The *tri5* deletion mutant still causes typical symptoms at the inoculated spikelets but fails to spread to neighboring spikelets on the same head via the rachis[9,10].

Like in *Fusarium sporotrichioides*, a producer of type A trichothecene T-2 toxin, the 15 *TRI* genes related to trichothecene biosynthesis are distributed on three chromosomes in *F. graminearum*[7,11–15].

[1]State Key Laboratory for Crop Stress Resistance and High-Efficiency Production, College of Plant Protection, Northwest A&F University, Yangling, Shaanxi 712100, China. [2]Department of Botany and Plant Pathology, Purdue University, West Lafayette, IN 47907, USA. ✉e-mail: jinrong@purdue.edu; cjiang@nwafu.edu.cn

Whereas *TRI1-TRI16* and *TRI101* are on chromosomes 1 and 3, respectively, the core *TRI* cluster that contains 12 genes, including *TRI5*, *TRI6*, and *TRI10*, is on chromosome 2[11,13]. The precursor for trichothecene synthesis is farnesyl pyrophosphate (FPP) which is synthesized via the mevalonate pathway[15–18]. The Tri5 trichodiene synthase catalyzes the first step in trichothecene biosynthesis by cyclizing FPP to trichodiene. The *TRI1*, *TRI3*, *TRI4*, *TRI8*, *TRI11*, and *TRI101* genes encode enzymes responsible for catalyzing the other steps of DON biosynthesis and their expression is under the control of *TRI6* and *TRI10*[7,18]. For the other *TRI* genes, *TRI12* encodes a major facilitator transporter functioning as the trichothecene efflux pump[19]. Whereas the function of *TRI9* and *TRI14* in trichothecene biosynthesis is not clear[20], *TRI7*, *TRI13*, and *TRI16* are pseudogenes with frameshift or nonsense mutations in DON-producing strains of *F. graminearum*[21–23].

In filamentous fungi, gene clusters responsible for the synthesis of secondary metabolites often contain one transcription factor that functions as a pathway-specific regulator of other genes in the same cluster. Interestingly, two of the *TRI* genes, *TRI6* and *TRI10*, function as transcriptional regulators of trichothecene biosynthesis in different *Fusarium* species[7,13]. In *F. graminearum*, both *TRI6* and *TRI10* are important for *TRI* gene expression, DON production, and plant infection[18,24]. Microarray analysis with infected wheat heads showed that the promoters of most *TRI* genes and other *F. graminearum* genes regulated by *TRI6* have the YNAGGCC motif[18], which is similar to the Tri6-binding site reported in *F. sporotrichioides*[25]. In a later study, a motif of GTGA repeats was identified by ChIP-seq analysis with a *TRI6*-HA transformant cultured under nutrient-rich conditions and verified to be recognized by Tri6 proteins in gel mobility shift assays[26]. Unlike Tri6 which is a $Cys_2His_2$ zinc finger protein, the Tri10 protein has no DNA-binding domain. Based on microarray analysis, *TRI6* expression is not affected in the *tri10* deletion mutant although deletion of *TRI6* has a minor effect on the expression of *TRI10*. Therefore, the mechanism for regulating *TRI* gene expression by *TRI10* and the functional relationship between *TRI6* and *TRI10* in regulating trichothecene biosynthesis are not clear in *F. graminearum*. Nevertheless, microarray analysis and qRT-PCR assays used in earlier studies could not distinguish sense and antisense transcripts and are not suitable to characterize the expression levels of *TRI* genes with antisense transcripts.

To better understand the roles of *TRI6* and *TRI10* in regulating *TRI* gene expression and DON biosynthesis, in this study, we conducted strand-specific RNA-seq analysis with the *tri6* and *tri10* deletion mutants under DON-inducing conditions. Although many *TRI* genes were down-regulated in both mutants, we found that *TRI10* regulated the expression of over 450 genes with diverse functions but *TRI6* was more specific for regulating *TRI* genes. As the master regulator, *TRI10* was essential for the expression of *TRI6* and suppression of antisense-*TRI6*. Antisense transcripts of *TRI5* and a lncRNA transcribed from its promoter (named RNA5P) also were identified in the *tri6* and *tri10* mutants. Although overexpression of RNA5P ectopically with the RP27 promoter had no effect, increased expression of RNA5P in situ resulted in a significant reduction or elimination of *TRI5* expression and DON production. In contrast, deletion of RNA5P increased DON production and *TRI5* expression. Taken together, these results indicate that both *TRI10* and *TRI6* are important for suppressing the expression of antisense-*TRI5* as well as RNA5P, which is a lncRNA inhibitory to *TRI5* expression and DON biosynthesis in *F. graminearum*.

## Results

### *TRI10* regulates more genes than *TRI6* under DON-inducing conditions

Both *TRI6* and *TRI10* are in the same transcription direction with *TRI5* and they are on a 7.5-kb region in the core *TRI* gene cluster (Fig. 1a). To identify genes regulated by *TRI6* and *TRI10*, we conducted RNA-seq analysis with the wild type and the *tri6* and *tri10* mutants[18] (Supplementary Table 1) cultured in DON-producing LTB (liquid trichothecene

biosynthesis) medium for three days. The *tri6* and *tri10* mutants we used were generated in a previous study by gene replacement with the hygromycin phosphotransferase (*hph*) cassette (Fig. 1a). Whereas the *TRI5-TRI6* region was 248-bp longer in the *tri6* mutant, the *TRI5-TRI10* region was 575-bp shorter in the *tri10* mutant in comparison with the wild type (Fig. 1a) due to the gene replacement events[18]. In comparison with the wild type, a total of 56 and 450 differentially expressed genes (DEGs) had over 2-fold changes in their expression levels in the *tri6* and *tri10* mutants, respectively. Among them, 40 and 16 DEGs were down- and up-regulated in the *tri6* mutant (Supplementary Data 1). In the *tri10* mutant, 231 and 219 DEGs were down- and up-regulated, respectively (Supplementary Data 2). These results indicate that *TRI10* regulates more genes than *TRI6* in *F. graminearum* in DON-producing cultures. GO enrichment analysis showed that DEGs down-regulated in the *tri6* mutant are enriched for genes involved in toxin metabolic process and pyridoxal phosphate biosynthesis or metabolism (Supplementary Fig. 1a). In the *tri10* mutant, down-regulated DEGs are enriched for genes associated with RNA transcription, binding, and processing (Supplementary Fig. 1b).

We then compared RNA-seq data of the *tri6* and *tri10* mutants and found that 17 and 14 DEGs were down- or up-regulated in both mutants (Fig. 1b and Supplementary Table 2). A total of 7 *TRI* genes were down-regulated in both mutants (Fig. 1c), indicating positive regulation by both *TRI6* and *TRI10*. Among the four *TRI* genes without significant changes in expression levels in the *tri6* or *tri10* mutant (Fig. 1c), *TRI7*, *TRI13*, and *TRI16* are pseudogenes with frameshift and nonsense mutations in DON-producing strains of *F. graminearum* such as PH-1[21–23]. For *TRI8* which encodes a trichothecene 3-O-esterase, its expression was reduced but less than 2-fold in the *tri6* or *tri10* mutant. Interestingly, among the 230 DEGs down-regulated in the *tri10* mutant, only *TRI4* was a DEG up-regulated in the *tri6* mutant (Fig. 1c). Because *TRI4* shares the same promoter region with *TRI6*, up-regulated expression of *TRI4* expression in the *tri6* mutant may be directly related to the abolishment of the self-inhibitory binding of Tri6 to its own promoter[26].

To evaluate the potential effects of *hph* cassette transcription on neighboring *TRI* genes, we generated the *tri6* and *tri10* deletion mutants with the *hph* cassette transcribed in the opposite direction from *TRI5* and *TRI6* or *TRI10* (Supplementary Fig. 2a, b). The resulting *tri6* and *tri10* deletion mutants had the same defects in *TRI5* expression and DON biosynthesis with the published mutants[18] (Supplementary Fig. 2c–e), indicating that the *TRI6* and *TRI10* gene deletion events but not the direction of *hph* are responsible for the defects observed in the *tri6* and *tri10* mutants.

### Expression of antisense transcripts of *TRI5*, *TRI6*, and *TRI11* in the *tri6* or *tri10* mutant

Because our data were generated by stranded RNA-seq, we then separated the sense and antisense transcripts of the *TRI* genes and examined for changes in their abundance in the *tri6* and *tri10* mutants compared to the wild type. In the wild type, only *TRI5* and *TRI6* had rare anti-sense transcripts (Supplementary Table 3). In the *tri10* mutant, antisense transcripts of *TRI5* and *TRI6* were significantly increased (Fig. 1d). For the *TRI5* gene, sense transcripts were rare but antisense transcripts were abundant in the *tri6* and *tri10* mutants (Fig. 1d), indicating a negative regulation of antisense-*TRI5* by *TRI6* and *TRI10*.

For *TRI11*, the only other *TRI* genes with antisense transcripts detected in this study, its sense transcripts were rare in the *tri6* and *tri10* mutants, a significant reduction compared to the wild type. Antisense transcripts of *TRI11* were present in both mutants but more abundant in the *tri10* mutant (Fig. 1d). These results indicate that *TRI6* and *TRI10* are not only essential for the expression of sense transcripts of *TRI5* and *TRI11* but also important for suppressing their antisense transcription. Based on the stranded RNA-seq data, antisense

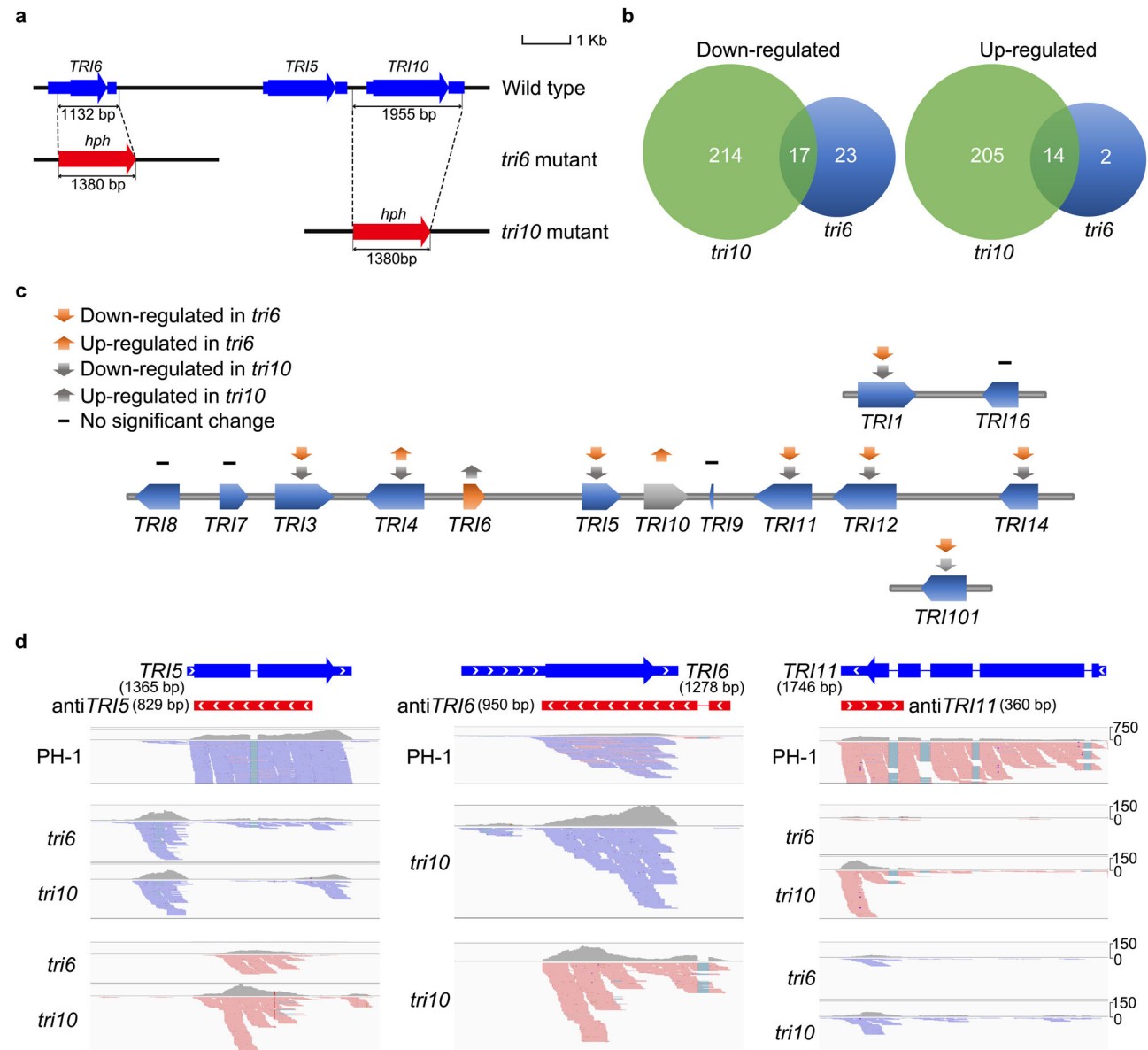

**Fig. 1 | Comparative analysis of DEGs identified in the *tri6* and *tri10* mutants.** **a** The structure of the core trichothecene gene cluster (*TRI6-TRI5-TRI10* locus) in the wild-type strain PH-1 and gene replacement events in the *tri6* and *tri10* deletion mutants. The transcription direction was indicated by the arrow. **b** Venn diagrams of DEGs down-regulated (left) and up-regulated (right) in the *tri6* and *tri10* mutants in comparison with the wild type. Among the 56 DEGs identified in the *tri6* mutant, 31 (over 55%) were also differentially expressed in the *tri10* mutant. **c** Schematic drawing of *TRI* genes and their expression profiles in the *tri6* and *tri10* mutants compared to the wild type. **d** IGV-Sashimi plots showing the read coverage and read counts of sense and anti-sense transcripts of *TRI5*, *TRI6*, and *TRI11* in the wild type and the *tri6* and *tri10* mutants. Schematic draws on the top showed their transcripts and ORFs. The scale on the right indicates the number of read counts. The length of sense and antisense transcripts of *TRI5*, *TRI6*, and *TRI11* were indicated in the figure.

transcripts of the other *TRI* genes were not detected in the wild type, and the *tri6* and *tri10* mutants (Supplementary Fig. 3).

### Repression of *TRI10* expression by *TRI6* and vice versa

*TRI10* has no antisense transcripts in the wild type and *tri6* mutant. However, the expression level of *TRI10* was significantly increased (over 2-fold) in the *tri6* mutant compared to the wild type (Fig. 2a). These results indicate that *TRI10* expression is negatively regulated by *TRI6*. In the *tri10* deletion mutant, the expression level of antisense-*TRI6* was significantly increased (over 4-fold) based on both RNA-seq (Fig. 1d) and qRT-PCR assays (Fig. 2b) compared to the wild type. For the sense transcripts of *TRI6*, the overall abundance was also increased in the *tri10* mutant due to a significant increase in partial transcripts mapped to the 3′-terminal region of *TRI6* (Fig. 1d; Fig. 2b). Close

examination showed that antisense transcripts of *TRI6* predominantly mapped to its 5′-terminal 410-bp region. In contrast, the 3′-terminal 274-bp region of *TRI6* was more abundant in its sense transcripts (Fig. 1d), suggesting the pairing and degradation of *TRI6* mRNA by antisense-*TRI6* transcripts. These results indicate that the expression of sense and antisense transcripts of *TRI6*, is negatively regulated by *TRI10*. Interestingly, antisense transcripts mapped to the terminator region of *TRI6* had a 65-bp intron (Fig. 1d) that is a part of antisense-*TRI6* based on RT-PCR assays (Supplementary Fig. 4a, b). To test the importance of this intron, we generated the *TRI6*^ME mutant allele, in which the GT-AG splicing sites were changed to AT-TG, and transformed into the *tri6* mutant (Supplementary Fig. 4a, b). In LTB cultures, the resulting *tri6/TRI6*^ME transformant was normal in DON biosynthesis and *TRI6* expression (Supplementary Fig. 4c, d), indicating that

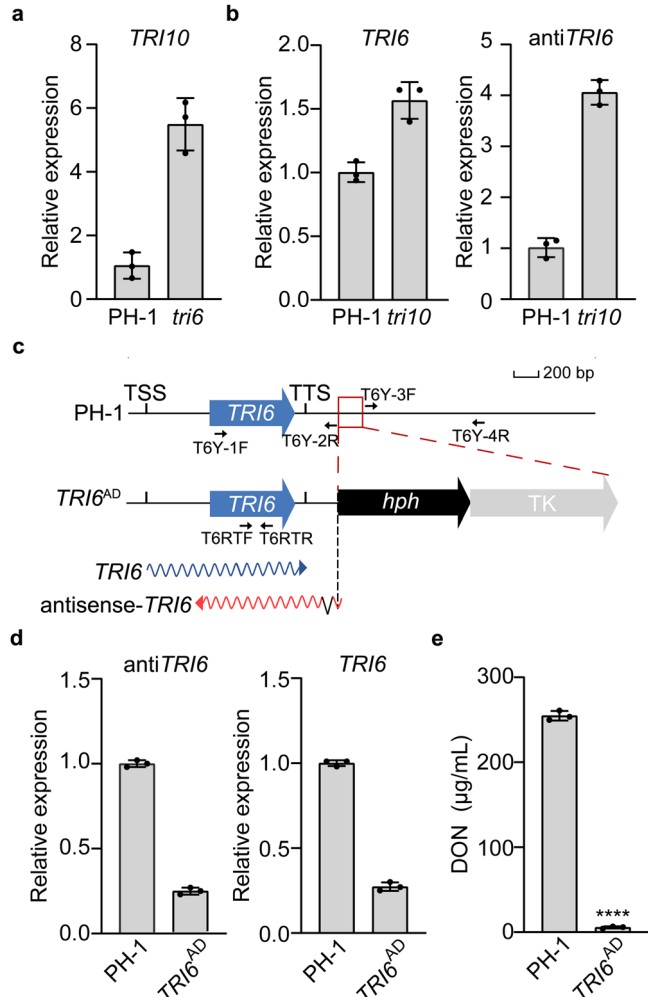

**Fig. 2 | Regulation of *TRI6* expression by *TRI10* and vice versa. a** Relative expression levels of *TRI10* in 3-day-old LTB cultures of the wild-type strain PH-1 (arbitrarily set to 1) and *tri6* mutant were assayed by qRT-PCR with primers amplified the 3'-terminal region of *TRI10*. **b** Relative expression levels of *TRI6* and antisense-*TRI6* in PH-1 (arbitrarily set to 1) and the *tri10* mutant were assayed with strand-specific qRT-PCR. Deletion of *TRI10* significantly increased the transcription of antisense-*TRI6*. **c** Schematic drawing of the *TRI6* ORF and its sense or antisense transcripts. In the *TRI6*^AD mutant allele, the 185-bp promoter region (red box) was replaced with the hygromycin phosphotransferase (*hph*) fusion with thymidine kinase (TK) genes. TSS/TTS, transcription start/termination site. Fragments were amplified with the primer pairs T6Y-1F/ T6Y-2R and T6Y-3F/T6Y-4R for the generation of the *TRI6*^AD strain, and the primer pair T6RTF/T6RTR was used for qRT-PCR assay. **d** Relative expression levels of *TRI6* and antisense-*TRI6* in 3-day-old LTB cultures of PH-1 and *TRI6*^AD mutant. **e** DON production in 7-day-old LTB cultures of PH-1 and the *TRI6*^AD mutants. For **a**, **b**, **d** and **e**, mean and standard deviation were estimated with data from three (*n* = 3) independent replicates (marked with black dots on the bars). For DON production, the statistical difference relative to PH-1 is based on the two-tailed unpaired *t* test (****, *p* < 0.0001). The exact *p*-values are shown in the Source Data file.

splicing of this intron in antisense transcripts is not important for the function of *TRI6*.

We then generated mutants deleted of a 185-bp region covering parts of the terminator region of *TRI6* by the gene replacement approach (Fig. 2c). This region likely contains the promoter region for antisense-*TRI6*. In the resulting *TRI6*^AD transformant (Supplementary Table 1), antisense transcripts of *TRI6* were significantly reduced in qRT-PCR assays with RNA isolated from LTB cultures (Fig. 2d). However, the expression of *TRI6* sense transcripts and DON production were also significantly reduced in the *TRI6*^AD mutant (Fig. 2d, e).

Therefore, deletion of this 185-bp region and insertion of the selectable marker at the *TRI6* locus likely affect the expression of both sense and antisense transcripts, possibly by interfering with the initiation of their transcription, transcription efficiency, or transcript stability.

### Overexpression of antisense-*TRI5* inhibits DON biosynthesis

Antisense transcripts of *TRI5* were not detectable in the wild type under DON inducing conditions but were abundant in both *tri6* and *tri10* mutants. To determine the function of antisense-*TRI5* transcripts, we generated the P$_{RP27}$-anti*TRI5* construct and transformed it into PH-1 (Fig. 3a). Transformants with the P$_{RP27}$-anti*TRI5* construct integrated ectopically were isolated (Supplementary Table 1) and assayed for *TRI5* and DON production in LTB cultures. In comparison with the wild type, the expression of antisense-*TRI5* was increased over 20-fold but DON production and the expression level of *TRI5* was significantly reduced in the P$_{RP27}$-anti*TRI5* transformants (Fig. 3b, c). In plant infection assays, P$_{RP27}$-anti*TRI5* transformants, like the *tri5* mutant, caused symptoms only on the inoculated wheat spikelet (Fig. 3d). These results indicate that overexpression of antisense-*TRI5* reduced the sense transcripts of *TRI5* and inhibitory to DON biosynthesis.

### RNA5P is a lncRNA mapped to the promoter region of *TRI5*

Close examination of strand-specific RNA-seq data showed that, under DON-inducing conditions, the 675-bp transcripts mapped to the promoter region of *TRI5* (named RNA5P for RNA transcribed from the *TRI5* promoter) were rare in PH-1 but became abundant in the *tri6* and *tri10* mutants, especially in the latter (Fig. 1c and Supplementary Table 3). Furthermore, RNA5P has one intron that was spliced in a small percentage of RNA5P transcripts in the *tri6* and *tri10* mutants (Fig. 1c). Alternative splicing of this intron will lead to two transcripts of RNA5P (Fig. 4a). Based on prediction with ExPASy (web.expasy.org/translate), these two transcripts may encode two small peptides of 58- and 86-amino acid residues that share the same stop codon (Fig. 4a and Supplementary Fig. 5). To test whether RNA5P is a part of *TRI5* mRNA, we conducted RT-PCR analysis with RNA isolated from LTB cultures of PH-1. Whereas primer pairs 5PRT1/5PRT2 and TF1/TR2 amplified the 0.11 and 0.18-kb band, respectively, no band could be amplified with primer pairs 5PRT1 and TR2 (Fig. 4b), indicating that RNA5P transcripts are independent of *TRI5* mRNA.

To test whether RNA5P transcripts can be translated, we generated the RNA5P-3×FLAG fusion construct (Fig. 4a) under the control of the constitutive RP27 promoter[27] and transformed it into PH-1. In the resulting P$_{RP27}$-RNA5P-3×FLAG transformants, the expression of RNA5P was increased in transformants with the fusion construct compared with PH-1 (Fig. 4c). However, the anti-FLAG antibody failed to detect any band on western blots of total proteins isolated from the resulting RNA5P-3×FLAG transformants cultured in LTB (Fig. 4d). As the control, the expression of FgEsa1 histone acetyltransferase was detected in the *FgESA1*-3xFLAG transformant[28] under the same experimental conditions (Fig. 4d). To further prove the function of RNA5P is independent of protein coding, we inserted an adenine (A) in the predicted ORF of RNA5P in situ (Fig. 4e and Supplementary Fig. 6). The resulting transformant with the frameshift mutation verified by sequencing analysis (Fig. 4f) was normal in *TRI5* expression (Fig. 4g) and DON production (Fig. 4h) in comparison with the wild type. These results suggest that RNA5P does not encode functional proteins and likely acts as a lncRNA in *F. graminearum*.

### Deletion of RNA5P by promoter switching increases *TRI5* expression

To characterize the function of RNA5P, we used the gene replacement approach to replace 834-bp region upstream from the *TRI5* ORF that contains the RNA5P region with an 834-bp fragment of the *TRI12* promoter (Fig. 5a, b). *TRI12* was selected because, like *TRI5*, its expression is regulated by both *TRI6* and *TRI10* but its expression

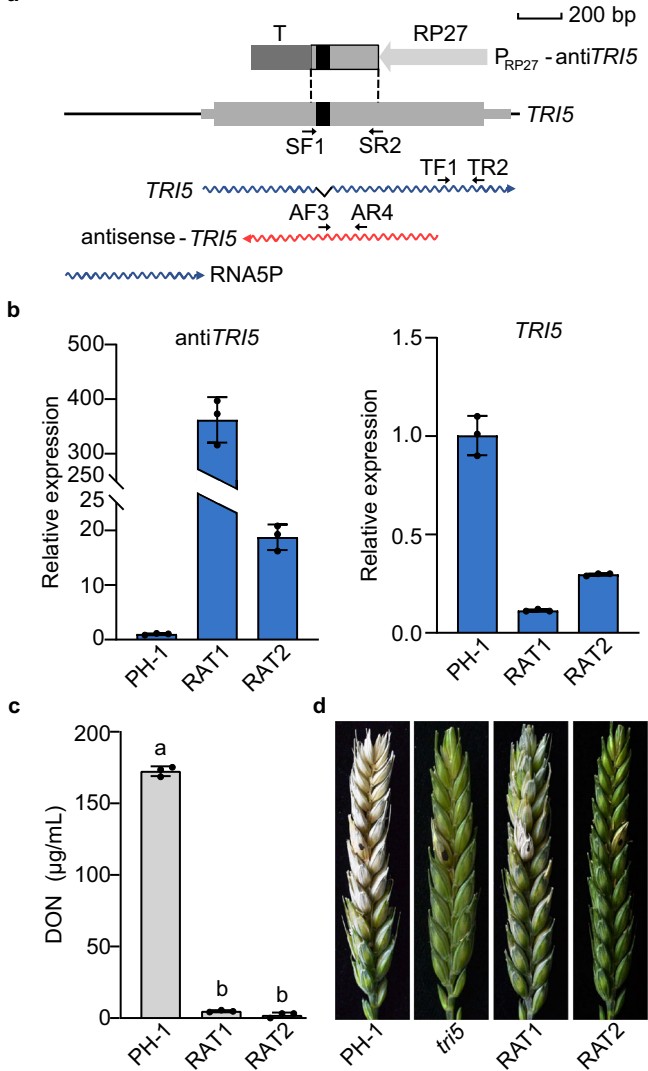

**Fig. 3 | Overexpression of antisense-*TRI5* inhibits DON biosynthesis.**
**a** Schematic drawing of the *TRI5* ORF, its sense or antisense transcripts, and lncRNA RNA5P as well as the P_RP27-anti*TRI5* construct in which a 299-bp fragment was cloned between the RP27 promoter (RP27) and CaMV ployA signal terminator (T). The black box marks the intron of *TRI5*. Arrows indicate the position and orientation of the primers. The fragment was amplified with the primer pair SF1/SR2 for the generation of the P_RP27-anti*TRI5* construct, and the primer pairs TF1/TR2 and AF3/AF4 were used to assay the expression of *TRI5* and antisense-*TRI5*, respectively.
**b** Relative expression levels of antisense-*TRI5* and *TRI5* in 3-day-old LTB cultures of the wild type (PH-1) and P_RP27-anti*TRI5* transformant (RAT1 and RAT2). The relative expression level in PH-1 was arbitrarily set to 1. **c** DON production in 7-day-old LTB cultures of PH-1 and P_RP27-anti*TRI5* (RAT1 and RAT2). **d** Wheat heads inoculated with the indicated strains were examined for head blight symptoms at 14 days post-inoculation (dpi). Black dots mark the inoculated spikelets. For **b** and **c**, mean and standard deviation were estimated with data from three (*n* = 3) independent replicates (marked with black dots on the bars). For DON production, different letters indicate significant differences based on the one-way ANOVA followed by Turkey's multiple range test. Differences were considered statistically significant when *p*-value is <0.05. The exact *p*-values are shown in the Source Data file.

level was 10-fold lower than that of *TRI5* based on our RNA-seq data (Fig. 5a, b). The size of the promoter regions used for promoter swapping was kept similar to avoid possible effects of size changes on local chromatin structures in the *TRI* gene cluster[14]. In the resulting P_TRI12-*TRI5* transformants, the replacement of *TRI5* promoter with that of *TRI12* was verified by PCR amplification (Supplementary Fig. 7).

When assayed by qRT-PCR with RNA isolated from LTB cultures, the expression of RNA5P was detected in the wild type but not in the P_TRI12-*TRI5* transformant (Fig. 5c). The expression level of *TRI5* sense transcripts was increased 2.6-fold in the P_TRI12-*TRI5* transformant compared to the wild type (Fig. 5c). However, the transcription of antisense-*TRI5* was only slightly increased in the P_TRI12-*TRI5* transformant (Fig. 5c). Consistent with increased expression of *TRI5*, DON production was also increased in the P_TRI12-*TRI5* transformant (Fig. 5d). Microscopical examination also showed a slight increase in the formation of hyphal swelling or cellular differentiation associated with DON production (Fig. 5e). These results indicate that even though *TRI12* has a lower expression level than *TRI5*, deletion of RNA5P by replacing the promoter of *TRI5* with that of *TRI12* had no significant effect on antisense-*TRI5* expression but significantly increased *TRI5* expression and DON biosynthesis.

## Overexpression of RNA5P ectopically has no inhibitory effect on *TRI5* expression and DON biosynthesis

To determine the effect of RNA5P overexpression, we generated the P_RP27-RNA5P construct (Fig. 6a) and transformed it into PH-1. In the resulting transformants (Supplementary Table 1) the transforming P_RP27-RNA5P construct was ectopically integrated into the wild-type strain PH-1 (Fig. 6a). The P_RP27-RNA5P transformants (LT7 and LT10) were normal in growth, conidiation, and sexual reproduction (Fig. 6b). In comparison with the wild type, the expression level of RNA5P was increased over 5-fold in LTB cultures in the P_RP27-RNA5P transformants (Fig. 6c). However, the expression level of *TRI5* (Fig. 6c) and DON production (Fig. 6d) were not significantly affected in P_RP27-RNA5P transformants. These results indicate that overexpression of RNA5P ectopically has no inhibitory effects on *TRI5* expression and DON biosynthesis. Therefore, the inhibitory effect of RNA5P on *TRI5* appears to be *cis*-acting and require its transcription from the *TRI5* locus.

## In situ overexpression of RNA5P with the TrpC promoter suppresses *TRI5* expression and DON production

To assay the effect of in situ RNA5P overexpression, a 366-bp fragment of the TrpC promoter was inserted 622-bp upstream from the transcription initiation site of RNA5P (126-bp downstream from the predicted Tri6-binding site) by homologous recombination (Fig. 7a). The integration of the TrpC promoter in the promoter region of RNA5P was verified by PCR amplification (Fig. 7a). The P_TrpC-RNA5P transformants (TR1 and TR2) were normal in growth (Fig. 7b). In LTB cultures, the expression level of RNA5P was almost 15-fold higher in the P_TrpC-RNA5P transformant than in the wild type (Fig. 7c), indicating increased expression of RNA5P by the TrpC promoter.

We then assayed *TRI5* expression and DON production in LTB cultures. In comparison with the wild type, the expression level of *TRI5* was significantly reduced in the P_TrpC-RNA5P transformants (Fig. 7c). DON production was almost non-detectable in the P_TrpC-RNA5P mutants (Fig. 7d). As a control, we also generated transformants with the TrpC promoter sequence integrated at the same site but in the opposite direction (Fig. 7a). Unlike the P_TrpC-RNA5P transformants, the resulting P_TrpC-inverted-RNA5P transformants (ATR1 and ATR3) were similar with the wild type in the expression levels of RNA5P and *TRI5* (Fig. 7c) as well as DON biosynthesis (Fig. 7d). These results indicate that in situ overexpression of RNA5P by the TrpC promoter negatively impacts *TRI5* expression and DON biosynthesis, confirming the *cis*-acting inhibitory effect of RNA5P.

## Elevated expression of RNA5P by the GTGA to TGAG mutation in its promoter also suppresses DON biosynthesis

To confirm the *cis*-acting effect of RNA5P, we introduced the GTGA (underlined) to TGAG mutation to the TCACGGGCTA-CAGTGAATGTTCGTGA sequence found at 748-bp upstream from its transcription initial site. This sequence contains two putative Tri6-

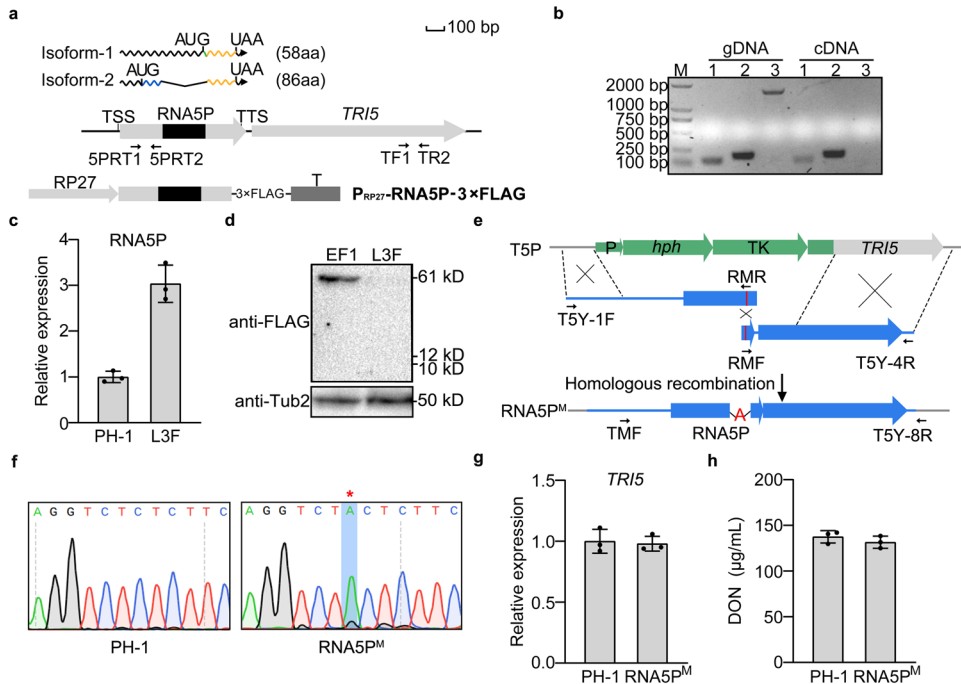

**Fig. 4 | RNA5P is a lncRNA transcribed from the *TRI5* promoter. a** Schematic drawing of the genomic region of RNA5P, its transcripts, and the $P_{RP27}$-RNA5P-3×FLAG fusion construct. Two isoforms of RNA5P (isoform-1 and isoform-2) are formed due to alternative splicing of the intron (black box) and are predicted to encode two small proteins that share the same stop codon. RP27, RP27 promoter; T, CaMV ployA signal terminator; TSS/TTS, transcription start/termination site. **b** PCR products amplified with primer pairs 5PRT1/5PRT2 (lane 1), TF1/TR2 (lane 2), and 5PRT1/TR2 (lane 3) from genomic DNA and 1st strand cDNA synthesized with RNA isolated from LTB cultures of the wild-type strain PH-1. M, molecular marker. The experiment was repeated three times independently with similar results. **c** Relative expression levels of RNA5P in LTB cultures of PH-1 and the $P_{RP27}$-RNA5P-3×FLAG transformant. **d** Western blots of total proteins isolated from transformants expressing the $P_{RP27}$-RNA5P-3×FLAG and the *FgESA1*-3×FLAG (as the positive control) fusion constructs were detected with the anti-FLAG and anti-Tubulin antibodies. **e** Schematic drawing of in situ nucleotide insertion in RNA5P. The *hph*-TK cassette in the *TRI5*$^{\Delta promoter}$ transformant (T5P) was replaced with a fragment with an adenine (A) insertion in the RNA5P (amplified with labeled primer pairs) by selecting transformants resistant to Floxuridine (RNA5P$^M$). P, TrpC promoter; T, CaMV ployA signal terminator. **f** The adenine (A) insertion in the RNA5P was verified by Sanger sequencing of the PCR products. The inserted nucleotide is marked with a red star. **g** Relative expression levels of *TRI5* were assayed by qRT-PCR with RNA isolated from LTB cultures of PH-1 (arbitrarily set to 1) and the RNA5P$^M$ strain. **h** DON production in 7-day-old LTB cultures of the PH-1 and the RNA5P$^M$ strain. For **c**, **g**, and **h**, mean and standard deviation were estimated with data from three (*n* = 3) independent replicates (marked with black dots on the bars). For DON production, no significant statistical differences were observed based on the two-tailed unpaired *t* test. The exact *p*-values are shown in the Source Data file.

binding sites[26] that overlap at the underlined GTGA (Fig. 8a). In the *TRI5*$^{M6B}$ transformants TM6 and TM66 in which the change of underlined GTGA to TGAG was confirmed by PCR amplification and sequencing (Fig. 8b), the expression level of RNA5P was significantly increased in the *TRI5*$^{M6B}$ transformants compared to the wild type in LTB cultures (Fig. 8c). In comparison with the wild type, the *TRI5*$^{M6B}$ transformants were significantly reduced in the expression level of *TRI5* (Fig. 8c) and DON production (Fig. 8d). These results suggest that like the self-inhibitory binding of Tri6 to its own promoter[26], binding of Tri6 to this GTGA repeat site is suppressive to the expression of RNA5P. Overexpression of RNA5P in situ by the GTGA to TGAG mutation that eliminates the inhibitory binding of Tri6 to its promoter is repressive to *TRI5* expression and DON production, confirming the *cis*-acting effect of RNA5P (Fig. 8e).

## Discussion

As an important virulence factor, DON biosynthesis plays a critical role in plant infection in *F. graminearum*. Similar to earlier studies[18], our RNA-seq data showed that both *TRI6* and *TRI10* are required for the expression of many *TRI* genes essential for DON biosynthesis. However, microarray analysis with the *tri6* and *tri10* mutants showed that *TRI10* is a pathway-specific regulator while *TRI6* functions as a regulator to regulate other genes functionally related to plant infection[18]. In this study, we found that deletion of *TRI6* only impacted the expression of 56 genes, which was significantly fewer than 450 DEGs in

the *tri10* mutant. One likely explanation for these contrasting observations is that, although the same strain was used, conditions for RNA isolate was different. For microarray analysis, RNA was isolated from infected wheat heads[18]. In this study, RNA used for RNA-seq analysis was isolated from LTB cultures. It is possible that *TRI6* regulates more genes related to plant infection but *TRI10* regulates many more genes than *TRI6* in DON-producing, axenic cultures. In *F. graminearum*, differences in *TRI* gene regulation exist between planta and cultures. In *F. graminearum*, differences in DON biosynthesis between axenic cultures and infected plant tissues have been observed in mutants deleted of *TRI14* that encodes a hypothetical protein of no known functions[20]. In *F. sporotrichioides*, *TRI10* functions as the principal regulatory element, orchestrating the expression of *TRI6*. While *TRI10* regulates many other genes, *TRI6* exhibits a more specific role in governing the *TRI* genes associated with T-2 toxin production[29,30], which is similar to what we observed in *F. graminearum*. Because sense transcripts are not distinguished from antisense transcripts by conventional microarray analysis and qRT-PCR assays, deletion of *TRI10* seemingly displayed no obvious effect on *TRI6* expression as shown by Seong and colleagues[18]. In this study, we found that *TRI6* is one of the three *TRI* genes with antisense transcripts. In comparison with the wild type, antisense transcripts of *TRI6* in the *tri10* mutant became more abundant as detected by microarray analysis and qRT-PCR assays[18]. For *TRI10*, it had no antisense transcripts, and its expression was increased over 5-fold in the *tri6* mutant, which is similar to the negative regulation of *TRI10*

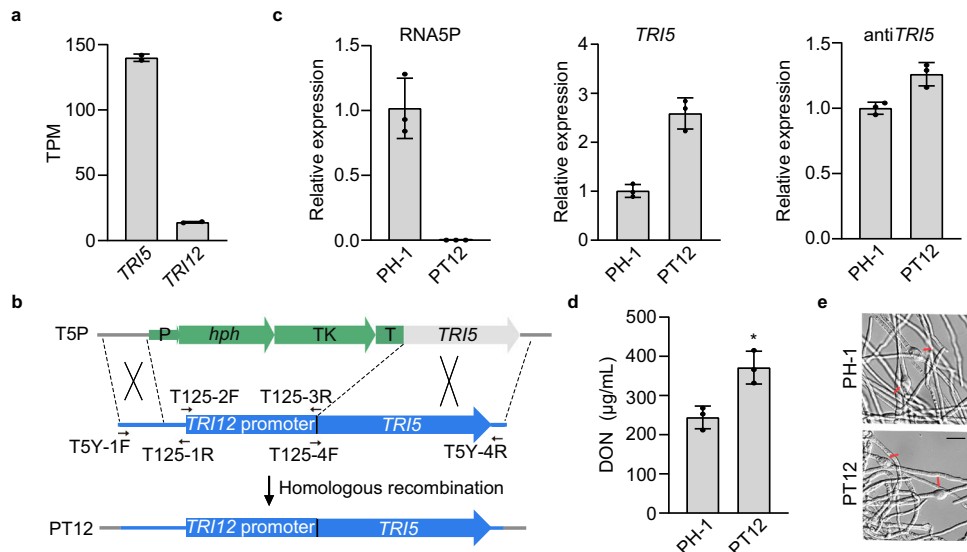

**Fig. 5 | Deletion of RNA5P increase *TRI5* expression and DON production.**
**a** Expression levels of *TRI5* and *TRI12* in the LTB cultures of the wild type based on RNA-seq data. **b** Schematic drawing of in situ replacement of the *TRI5* promoter with the *TRI12* promoter. The hygromycin phosphotransferase (*hph*)-thymidine kinase (TK) cassette in the *TRI5*^Δpromoter transformant (T5P) was replaced with the *TRI12* promoter (amplified with labeled primer pairs) by selecting transformants resistant to Floxuridine (PT12). P, TrpC promoter; T, CaMV ployA signal terminator. **c** Relative expression levels of RNA5P, *TRI5*, and antisense-*TRI5* were assayed by

qRT-PCR with RNA isolated from LTB cultures of PH-1 (arbitrarily set to 1) and the P$_{TRI12}$-*TRI5* transformant PT12. **d** DON production in 7-day-old LTB cultures of the marked strains. **e** LTB cultures of PH-1 and P$_{TRI12}$-*TRI5* transformant PT12 were examined for bulbous structures (marked with arrows) related to DON production. Bar = 20 μm. For **a**, **c**, and **d**, mean and standard deviation were estimated with data from three (*n* = 3) independent replicates (marked with black dots on the bars). For DON production, the statistical difference relative to PH-1 is based on the two-tailed unpaired *t* test (*, *p* < 0.05). The exact *p*-values are shown in the Source Data file.

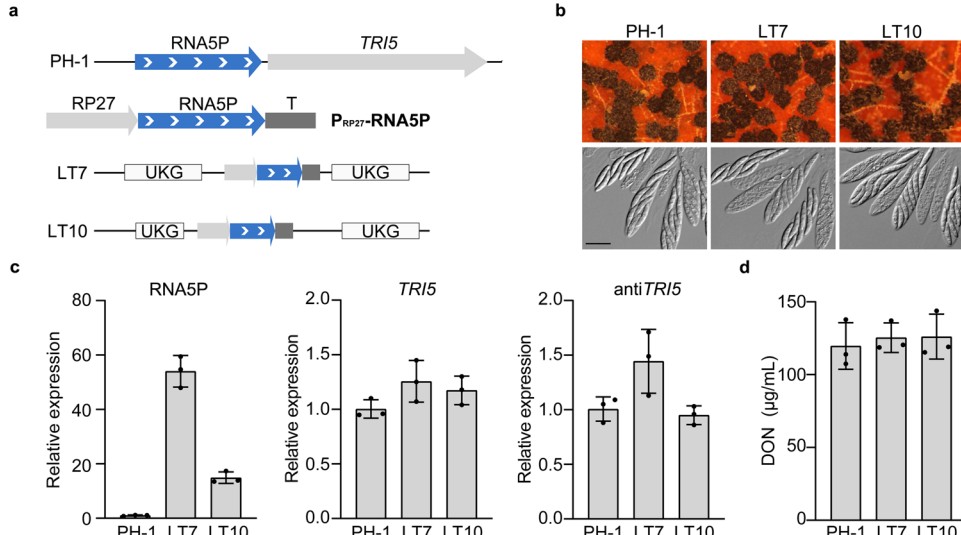

**Fig. 6 | Overexpression of RNA5P has no effect on *TRI5* expression and DON production. a** Schematic drawing of RNA5P upstream of *TRI5* and the generation of P$_{RP27}$-RNA5P construct. RP27, PR27 promoter; T, CaMV polyA signal terminator. LT7 and LT10 are two independent transformants with the integration of P$_{RP27}$-RNA5P ectopically. UKG, Unknown gene. **b** Mating cultures of PH-1 and P$_{RP27}$-RNA5P transformants (LT7 and LT10) were examined for perithecium formation at 7 days post-fertilization (dpf). Asci and ascospores were examined by DIC microscopy. Bar = 20 μm. **c** Relative expression levels of RNA5P, *TRI5*, and antisense-*TRI5* were

assayed by qRT-PCR with RNA isolated from LTB cultures of PH-1 (arbitrarily set to 1) and P$_{RP27}$-RNA5P transformants (LT7 and LT10). **d** DON production in 7-day-old LTB cultures of the same set of strains. For **c** and **d**, mean and standard deviation were estimated with data from three (*n* = 3) independent replicates (marked with black dots on the bars). For DON production, no significant statistical differences were observed based on the one-way ANOVA followed by Turkey's multiple range test. Differences were considered statistically significant when *p*-value is <0.05. The exact *p*-values are shown in the Source Data file.

by *TRI6* in *F. sporotrichioides*[29]. Therefore, *TRI6* and *TRI10* may regulate each other's expression in LTB cultures with *TRI10* is the master, global regulator in *F. graminearum*. Because *TRI6* expression is regulated by *TRI10* and lacks any known DNA-binding domain, it is possible that regulation of other *TRI* genes by *TRI10* is mediated by *TRI6*. In concordance, *TRI6* was reported to interact with pathway-specific

transcription factor Gra2 for the regulation of gramillin biosynthesis[31]. *TRI10* may also regulate different subsets of genes by interacting with pathway-specific transcription factors.

Interestingly, a recent study with the *Tri6_nsm* mutant carrying a nonsense mutation in *TRI6* (no changes in the length of *TRI6*) and Δ*Tri6* tk gene replacement mutant (with an increase of 7.0-kp at the *TRI6*

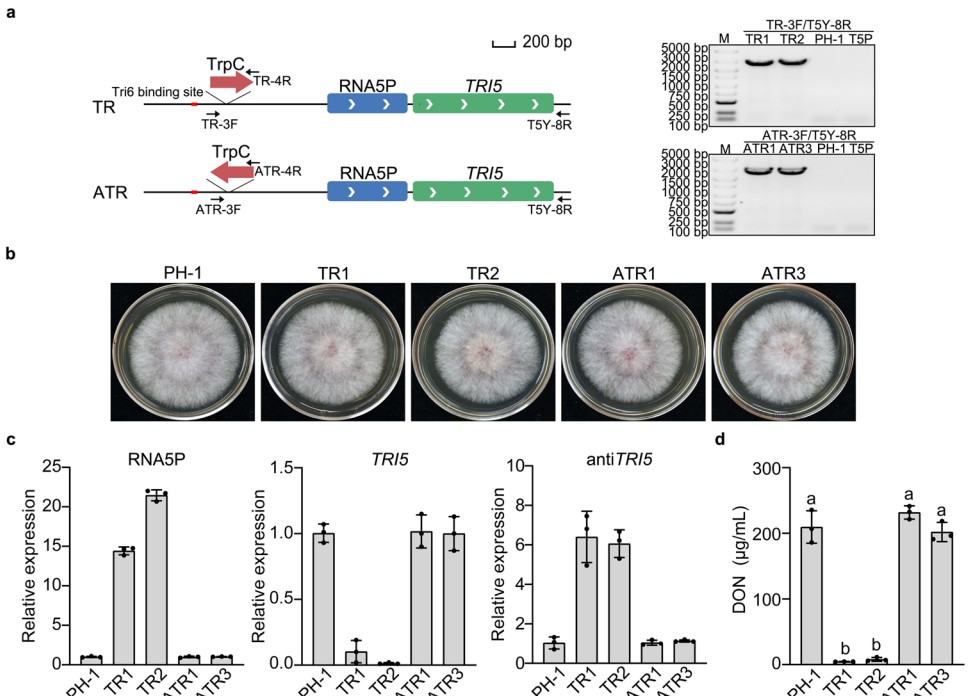

**Fig. 7 | In situ overexpression of RNA5P with the TrpC promoter inhibits *TRI5* expression. a** Insertion of the TrpC promoter in the forward and reverse directions upstream from RNA5P. Red box marks the Tri6-binding site (left panel). PCR products amplified with primer pairs TR-3F/T5Y-8R and ATR-3F/T5Y-8R from genomic DNA of the wild-type strain PH-1, *TRI5*^Δpromoter transformant (T5P), and the resulting transformants (right panel). The TrpC promoter was successfully inserted upstream of the RNA5P to generate the resulting transformants TR1 and TR2, and the insertion of TrpC-inverted promoter upstream of the RNA5P was verified by PCR in the transformants ATR1 and ATR3. Genomic DNA isolated from the wild-type PH-1 and T5P was used as a control. M, molecular marker. The experiment was repeated three times independently with similar results. **b** Three days PDA cultures of the wild type (PH-1), P$_{TrpC}$-RNA5P mutants (TR1 and TR2), and P$_{TrpC-inverted}$ transformants (ATR1 and ATR3). **c** Relative expression levels of RNA5P, *TRI5*, and antisense-*TRI5* were assayed by qRT-PCR with RNA samples isolated from LTB cultures of the wild type (PH-1), P$_{TrpC}$-RNA5P mutants (TR1 and TR2), and P$_{TrpC}$-Inverted-RNA5P transformants (ATR1 and ATR3). The relative expression level in PH-1 was arbitrarily set to 1. **d** DON production in LTB cultures of the same set of strains. For **c** and **d**, mean and standard deviation were estimated with data from three ($n = 3$) independent replicates (marked with black dots on the bars). For DON production, different letters indicate significant differences based on one-way ANOVA followed by Turkey's multiple-range test. Differences were considered statistically significant when *p*-value is <0.05. The exact *p*-values are shown in the Source Data file.

locus) showed that the expression of *TRI10* was elevated in the latter but not in the *Tri6_nsm* mutant[14], suggesting a possible effect of changes in the length of *TRI6* and regional chromatin structures on *TRI10* expression. Unlike the 7.0-kb increase the Δ*Tri6* tk strain, the gene replacement event in the *tri6* mutant resulted an increase of only 248-bp at the *TRI6* locus[18], which is 2.6-kp upstream from *TRI5* (Fig. 1a). The defect of the *tri6* mutant in DON biosynthesis was largely complemented by ectopic expression of *TRI6*[18] confirming that the phenotype observed in the mutant is related to *TRI6* deletion. Similarly, the *tri10* deletion mutant was complemented by ectopic expression of *TRI10* and the gene replacement event at this locus resulted in a shortening of 575-bp[18]. Furthermore, changes in chromatin structures usually create an environment that is permissive for similar activation or repression of neighboring genes[32], however, the *tri6* deletion mutant was significantly reduced in *TRI5* expression but increased in *TRI10* expression although all three genes in the *TRI6-TRI5-TRI10* region are transcribed in the same direction (Fig. 1a). The opposite effects of *TRI6* deletion on the expression of *TRI5* and *TRI10* further indicate that changes in *TRI* expression in the *tri6* mutant is likely due to its transcriptional regulation by *TRI6*. Nevertheless, we noticed differences in culture conditions and *F. graminearum* strains used by Liew and colleagues[14] and in this study[18]. Because DON is a secondary metabolite, we normally assay DON production and *TRI* gene expression in cultures of strain PH-1[33] that have been incubated for at least for 72 h. It is also noteworthy that the nonsense mutation at the Methionine (Met) residue (A^10TG^12 to TAG) in the *Tri6_nsm* mutant of strain

JCM 9873[14] is towards the N-terminus of Tri6 and truncated proteins may be translated from downstream in-frame ATG or CTG/GTG codons[34]. Overall, the regulation on *TRI* genes is multilayered and involves transcription factors, antisense transcripts, and chromosomal organization.

In comparison with the wild type, the expression of antisense-*TRI5* was significantly up-regulated in the *tri6* and *tri10* mutants, suggesting a negative regulation by *TRI6* and *TRI10*. Because full-length sense transcripts of *TRI5* were rare or absent in the *tri6* and *tri10* mutants, detection of *TRI5* expression by qRT-PCR or microarray analysis in both mutants in earlier studies[18] could be due to its antisense transcripts. For the RNA5P lncRNA transcribed from the promoter region of *TRI5*, its expression is also negatively regulated by both *TRI6* and *TRI10*. Because mutations at the GTGA repeats in its promoter increased RNA5P expression, *TRI6* may represses RNA5P expression by inhibitory binding to this predicted Tri6-binding site. For *TRI10*, it may repress RNA5P expression via its regulation of *TRI6*. However, repression by inhibitory binding of Tri6 does not explain the low expression level of RNA5P in the wild type under DON producing conditions. Similar to antisense-*TRI5*, a low-level expression of RNA5P in the wild type may be related to fine adjustment of trichodiene synthase activities to avoid excessive DON or DON toxicity in *F. graminearum*.

Although lncRNAs have a broad range of mode of actions[35,36], transcriptional regulation of neighboring protein-coding genes is one of the common mechanisms for lncRNAs[37]. In this study, we showed that deletion of RNA5P by replacing the *TRI5* promoter with that of

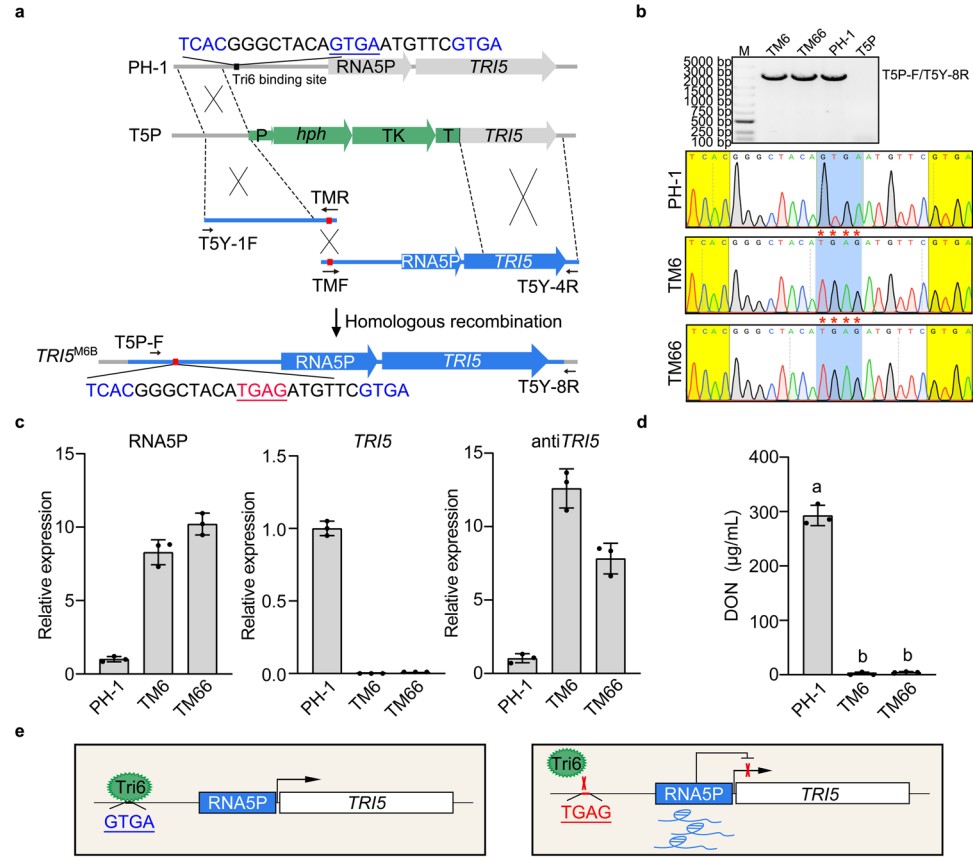

**Fig. 8 | The GTGA to TGAG mutation in the promoter of RNA5P affects its expression and DON biosynthesis. a** Schematic drawing of the genomic region of RNA5P and the gene replacement event to introduce the GTGA (underlined) to TGAG mutation at the Tri6-binding site in its promoter. The overlapping PCR products amplified with labeled PCR primers containing the TCACGGGCTA-CATGAGATGTTCGTGA sequence were transformed into the *TRI5*^Δpromoter transformant (T5P) to generate the *TRI5*^M6B transformants carrying the GTGA to TGAG mutation. P, TrpC promoter; T, CaMV ployA signal terminator. **b** The expected sizes of fragments could be amplified with primer pairs T5P-F and T5Y-8R from genomic DNA of the wild type and two independent *TRI5*^M6B transformants (TM6 and TM66), but not the T5P strain (upper panel). M, molecular marker. The GTGA-to-TGAG mutation in TM6 and TM66 strains was verified by Sanger sequencing of the PCR products (lower panel). The mutations are marked with red stars. The experiment was repeated three times independently with similar results. **c** Relative expression

levels of RNA5P, *TRI5*, and antisense-*TRI5* in LTB cultures of PH-1 (arbitrarily set to 1) and two independent *TRI5*^M6B transformants (TM6 and TM66) carrying the GTGA to TGAG mutation. **d** DON production in LTB cultures of PH-1 and two independent *TRI5*^M6B transformants (TM6 and TM66). **e** A proposed model for the role of Tri6-binding on RNA5P expression. In the wild type, binding of Tri6 is inhibitory to RNA5P expression, leading to *TRI5* expression under DON-producing conditions. In the mutant with the GTGA-to-TGAG mutation that eliminates Tri6-binding, elevated RNA5P expression results in the suppression of *TRI5* expression. For **c** and **d**, mean and standard deviation were estimated with data from three ($n = 3$) independent replicates (marked with black dots on the bars). For DON production, different letters indicate significant differences based on one-way ANOVA followed by Turkey's multiple-range test. Differences were considered statistically significant when $p$-value is <0.05. The exact $p$-values are shown in the Source Data file.

*TRI12* resulted in a significant increase in *TRI5* expression and DON production although *TRI12* has a lower expression level than *TRI5*. Furthermore, we showed that overexpression RNA5P ectopically with the RP27 promoter had no effect but overexpression of RNA5P in situ with the TrpC promoter reduced *TRI5* expression and DON production. In addition, when the Tri6-binding site in the RNA5P promoter region was mutagenized, the expression of RNA5P was significantly increased but *TRI5* expression and DON production were suppressed. These results indicated that RNA5P has a *cis*-acting inhibitory effect on the transcription of *TRI5*, which is essential for DON biosynthesis. *TRI6* may regulate the expression of *TRI5* by binding with the RNA5P promoter region upstream of *TRI5*. The putative Tri6-binding site (GTGA/TCAC repeats separated by 0 to 8 nucleotides) is present in the promoter region of *TRI5* in both *F. culmorum* and *F. sporotrichioides* although at different positions.

To our knowledge, RNA5P is the first lncRNA known to regulate secondary metabolism in filamentous fungi. In yeast, plants, and animals, some *cis*-acting lncRNAs mediate gene silencing through the recruitment of polycomb repressive complex 2 (PRC2), while others

repress the transcription of target genes by nucleosome rearrangements or formation of RNA-DNA[38,39]. In *F. graminearum*, RNA5P is transcribed from the promoter region of *TRI5*. The expression of RNA5P lncRNA may inhibit the transcription of *TRI5* by causing repositions of nucleosomes or the formation of RNA-DNA triplexes at its transcription initiation sites. It is also possible that RNA5P suppresses *TRI5* expression by recruiting PRC2 known to be important for repressing *TRI* genes and other secondary metabolite gene clusters[40] to the promoter region of *TRI5*. Therefore, further characterizations are necessary to determine the molecular mechanism of *cis*-acting regulation of *TRI5* expression by RNA5P in *F. graminearum*.

## Methods

### Strains and cultural conditions

The wild type and mutant strains[33] and transformants generated in this study (Supplementary Table 1) were routinely cultured on potato dextrose agar (PDA) plates. PDA cultures incubated at 25 °C for three days were used to examine growth rate and colony morphology[41]. Conidiation and conidium morphology were assayed with conidia

harvested from 5-day-old carboxymethyl cellulose (CMC) cultures[42]. To assess defects in sexual reproduction, aerial hyphae of 5-day-old carrot agar cultures were pressed down with sterile 0.1% Tween 20 and then incubated at 25 °C under black light. Perithecium formation was examined 7 days after induction for sexual reproduction[43]. PEG-mediated transformation of protoplasts was performed for gene replacement[42]. Hygromycin B (H005, MDbio, China) and geneticin (345810, Sigma-Aldrich, USA) were added to the final concentration of 300 and 150 μg/ml, respectively, for transformation selection.

### Plant infection and DON production assays

Conidia were collected from 5-day-old CMC cultures and resuspended to a final concentration of $10^5$ spores/ml in sterile distilled water. For each wheat head of cultivar Xiaoyan 22, the fifth spikelet from the bottom was drop-inoculated with 10 μl of conidium suspension. Wheat heads that underwent inoculation were covered with plastic bags for 48 hours to maintain moisture. Subsequently, the examination of infected wheat heads for diseased spikelets was conducted at 14 days post-inoculation (dpi) to determine the disease index, quantifying the number of diseased spikelets per head[44,45]. For assaying DON production, conidia were resuspended to a final concentration of $10^4$ conidia/ml in LTB medium[46]. After incubation for 7 days, DON was measured by GCMS-QP2010 system with AOC-20i autoinjector (Shimadzu Co. Japan)[47,48]. For each strain, mean and standard deviation were calculated with data from three biological replicates.

### Generation of the *TRI6*[AD], *TRI5*[Δpromoter], *tri6*, *tri10* mutant alleles and transformants

To generate the *TRI6*[AD] allele deleted of the 185-bp putative promoter region of antisense-*TRI6* with the split marker approach[49], 1-kb each of its upstream and downstream fragments was amplified with the primer pairs indicated in Fig. 2c and ligated to the 5′- and 3′- terminal regions of the hygromycin phosphotransferase (*hph*) cassette fused with the thymidine kinase (TK) by overlapping PCR. The fused *hph*-TK selectable marker is suitable for selection for resistance against hygromycin and sensitivity to 5-fluoro-2′-deoxyuridine (Floxuridine) in fungi[50,51]. The resulting PCR products were transformed into protoplasts of PH-1 to isolate transformants resistant to hygromycin and screen for *TRI6*[AD] mutants by PCR for the deletion of the 185-bp region of *TRI6*. Similar approaches were used to generate transformants deleted of the promoter of *TRI5* (*TRI5*[Δpromoter]) as indicated in Supplementary Fig. 6.

To generate the *tri6* mutant (AT6) by replacing the *TRI6* gene with a reverse-transcribed *hph* cassette, its upstream and downstream fragments were amplified with the primer pairs indicated in Supplementary Fig. 2 and ligated to the 3′- and 5′- terminal regions of the hygromycin phosphotransferase (*hph*) cassette by overlapping PCR. The resulting PCR products were transformed into protoplasts of PH-1 to isolate transformants resistant to hygromycin and screen for the *tri6* mutants by PCR. Similar approaches were used to generate the *tri10* deletion mutant (AT10). All the primers used are listed in Supplementary Data 3.

### Generation of the *TRI6*[ME], *TRI5*[M6B] and RNA5P[M] mutants

To generate the *TRI6*[ME] allele, the GT-AG to AT-TG mutations were introduced by overlapping PCR with mutations in the primers. The resulting mutant allele were fused with flanking sequences of *TRI6* amplified with primer pairs T6Y-1F/MER and MEF/T6Y-4R and transformed into protoplasts of *TRI6*[AD] transformant. Transformants resistant to 25 μg/ml floxuridine (HY-B0097, MCE, USA) were isolated and screened by PCR for the replacement of *hph*-TK with the *TRI6*[ME] allele by homologous recombination[50,51]. For *TRI5*[M6B], the GTGA to TGAG mutation at −1482 to −1479 was introduced by overlapping PCR with mutations in the primer pairs indicated in Fig. 8a. The resulting *TRI5*[M6B] fragment was then fused with flanking sequences of *TRI5* and transformed into protoplasts of the *TRI5*[Δpromoter] mutant T5P. Floxuridine-

resistant transformants were isolated and screened for the replacement of *hph*-TK with *TRI5*[M6B]. Similar approaches were used to generate the RNA5P[M] transformant which has an adenine (A) insertion in the RNA5P as indicated in Fig. 4e. All the primers used were listed in Supplementary Data 3.

### Generation of P[TRI12]-*TRI5*, P[TrpC]-RNA5P and P[TrpC-inverted]-RNA5P transformants

To replace the promoter of *TRI5* with that of *TRI12*, a fragment containing the *TRI12* promoter region was amplified with primer pairs indicated in Fig. 5b and fused to the upstream or downstream flanking sequences of *TRI5* by overlapping PCR. The resulting PCR products were transformed into protoplasts of the *TRI5*[Δpromoter] mutant T5P. Transformants resistant to floxuridine were isolated and screened by PCR for the replacement of *hph*-TK with *TRI5* promoter. To generate the P[TrpC]-RNA5P transformants, the TrpC promoter was fused with the upstream and downstream flanking sequences of RNA5P amplified with primer pairs T5Y-1F/TR-2R, TR-5F/T5Y-4R by overlapping PCR. The resulting PCR products were transformed into the *TRI5*[Δpromoter] mutant T5P. Transformants resistant to floxuridine were isolated and verified for the integration of P[TrpC]-RNA5P at *TRI5*. Similar approaches were used to generate P[TrpC-inverted]-RNA5P transformant in which the TrpC promoter is in the opposite direction of RNA5P transcription. All the primers used were listed in Supplementary Data 3.

### Generation of P[RP27]-RNA5P and P[RP27]-antisense-*TRI5* constructs and transformants

The yeast gap repair approach[52] was used to generate the andP[RP27]-RNA5P and P[RP27]-antisense-*TRI5* constructs. For RNA5P, it was amplified with primer pair LT1F/LT2R and fused with the CaMV ployA signal terminator[53] amplified with primer pair LT3F/LT4R. The resulting PCR product was co-transformed with *XhoI*-digested pFL2 (geneticin resistance) into yeast strain XK1-25[54]. The recombined P[RP27]-RNA5P construct was rescued from Trp+ transformants and transformed into PH-1 after verification by sequencing. Transformants resistant to geneticin were isolated and verified by PCR for the integration of P[RP27]-RNA5P ectopically.

For antisense-*TRI5*, a 299-bp fragment corresponding to 449-747 bp of the *TRI5* sense transcript was amplified with primer pair SF1/SR2. The resulting PCR product was closed into pFL2 by gap repair as outlined above. The P[RP27]-RNA5P construct was verified by sequence analysis and transformed into protoplasts of PH-1. Geneticin-resistant transformants were isolated and verified by PCR for the integration of P[RP27]-antisense-*TRI5* ectopically. All the primers used were listed in Supplementary Data 3.

### Stranded RNA-seq analysis

Total RNAs were isolated from hyphae harvested from 3-day-old LTB cultures with the TRIzol reagent for RNA-seq analysis (15596018, Invitrogen, USA). Conidia used to inoculate LTB were collected from 5-day-old CMC cultures. Strand-specific RNA-seq libraries constructed with the NEB Next Ultra Directional RNA Library Prep Kit (E7765, NEB, USA) were sequenced with Illumina HiSeq 2500 at Novogene Bioinformatics Technology (China). The resulting RNA-seq reads were mapped onto the updated version (Named YL1) of the reference genome of *F. graminearum* strain PH-1[55] by HISAT2[56] and visualized with the Integrative Genomics Viewer (IGV) tool (software.broadinstitute.org/software/igv/)[55]. The number of reads (count) mapped to each gene was calculated by feature Counts[57]. Genes with $\log_2$FC greater than 1 and FDR less than 0.05[58] were identified as Differentially Expressed Genes (DEGs) using the edge Run package with the exact Test function. GO enrichment analysis was analyzed with Blast2GO[59]. Data from two independent biological replicates were used for differential expression analysis. The RNA-seq data generated in this study have been deposited in the NCBI Sequence Read Archive database under accession

code PRJNA1044545. The reference genome of *F. graminearum* strain PH-1[55] used in this study is available in the NCBI GenBank database under accession code PRJNA782099.

## Quantitative reverse transcription-polymerase chain reaction (qRT-PCR) assays

Total RNA of each strain was isolated from 3-day-old LTB cultures with TRIzol for qRT-PCR assay (Invitrogen, USA). Conidia used to inoculate LTB were collected from 5-day-old CMC cultures. First-strand cDNA was synthesized with Fast Quant RT Kit (KP1116, TIANGEN, China) before performing qRT-PCR assays with the CFX96 Real-Time System (Bio-RAD, USA)[60]. Primer pairs for amplifying RNA5P, *TRI5*, antisense-*TRI5*, *TRI6*, antisense-*TRI6*, *TRI10*, and *TRI12* are 5PRT1/5RPT2, TF1/TR2, AF3/AF4, T6RTF/T6RTR (for both *TRI6* and antisense-*TRI6*), T10RTF/T10RTR and T12RTF/T12RTR, respectively. The *ACTIN* gene (FG4G14550) was used as the internal control[28]. The relative expression level of individual genes or transcripts was calculated by the $2^{-\Delta\Delta Ct}$ method[61]. Data from three independent biological replicates were used to calculate the mean and standard deviation. All the primers used were listed in Supplementary Data 3.

## Western blot analysis with the RNA5P-3×FLAG transformants

To generate the $P_{RP27}$-RNA5P-3×FLAG-$T_{CaMV}$ construct by yeast gap repair[52], the RNA5P-3×FLAG fusion construct was generated by PCR amplification with primers pair LT1F/L3F-R and fused with the CaMV ployA terminator[53] by overlapping PCR. The resulting PCR product was co-transformed with *Xho*I-digested pFL2 into yeast strain XK1-25[54]. The recombined $P_{RP27}$-RNA5P-3×FLAG-$T_{CaMV}$ construct was rescued from Trp+ transformants and transformed into PH-1 after verification by sequencing. Transformants resistant to geneticin were screened by PCR. Total proteins were isolated from the resulting transformants with hyphae harvested from 3-day-old LTB cultures. For western blot analyses, total proteins were separated on 12.5% SDS-PAGE gels and transferred to nitrocellulose membranes[54]. The primary anti-FLAG antibody (F9291, Sigma, USA) and secondary anti-Mouse antibody (DY60203, DEEYEE, China) were used to detect the expression of RNA5P-3×FLAG fusion proteins. Detection with a primary anti-Tub2 β-tubulin antibody (HA720035, HUABIO, China)[62] and secondary anti-Rabbit antibody (DY60202, DEEYEE, China) were used as the loading control. Proteins isolated from the *ESA1*−3×FLAG transformant[28] were used as the positive control for detection with the anti-FLAG antibody. A dilution of 1:1,000 is used for primary antibodies and 1:10,000 for secondary antibodies.

## Reporting summary

Further information on research design is available in the Nature Portfolio Reporting Summary linked to this article.

# Data availability

Data supporting the findings of this work are available within the paper and its Supplementary Information files. The RNA-seq data generated in this study have been deposited in the NCBI Sequence Read Archive database under accession code PRJNA1044545. The reference genome of *F. graminearum* strain PH-1 used in this study is available in the NCBI GenBank database under accession code PRJNA782099. Source data are provided with this paper.

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

## Acknowledgements

We thank Zhe Tang, Ping Xiang, Yuhan Zhang, Xinyu Cao, Xinlong Gao for assistance with sample preparation, DON measurement, and microscopic observation. We also thank Drs. Qinhu Wang, Guanghui Wang, and Ming Xu for fruitful discussions. This work was supported by grants from the National Key R&D Program of China (2022YFD1400100),

Shaanxi Science Fund for Distinguished Young Scholars (2022JC-14), the National Youth Talent Support Program, and Innovation Capability Support Program of Shaanxi (No. 2023-CX-TD-56) to CJ as well as a grant from the USWBSI to JRX.

## Author contributions

J.R.X., C.J., and P.H. conceived and designed the experiments; P.H., X.Y., M.D., and Z.W. performed the experiments; P.H., H.L., and C.J. contributed materials/analysis tools and analyzed the data; J.R.X., C.J., and P.H. wrote the paper.

## Competing interests

The authors declare no competing interests.
