## [Peer Review File · Nature Communications]

Regulation of TRI5 expression and deoxynivalenol biosynthesis by a long non-coding RNA in *Fusarium graminearum*Reviewer #1 (Remarks to the Author):

Please see the uploaded PDF file (Comments to authors).

Reviewer #1 Attachment on the following page

This is an interesting paper introducing a new model of trichothecene biosynthesis regulation by Tri6p. By mutating the 4-bp Nasmith sequence (GTGA repeats, separated by 8 bp or less) at its native locus, the authors showed positive regulation of *TRI5* through repression of the RNA5P transcript. Although my group think of a quite different model of regulatory mechanism of toxin production, the authors' model (**Fig. 8d**: which assumes suppression of RNA5P transcription by binding of Tri6p to the Nasmith sequence) has a large impact and could stimulate research progress and discussion in the relevant community. Although other reviewers may request EMSA data as Nasmith et al. (ref. 22) did so, I do not think them necessary in this paper. Indeed, full-length Tri6-HA recombinant protein cannot be easily obtained in an active form, as described in ref. 22 (page 10, left column last paragraph of ref. 22), and we had experienced difficulties in obtaining active form of recombinant Tri6p in *E. coli*. Provided that the molecular genetic data on the role of the Nasmith sequence are satisfactorily presented in this manuscript, I think the authors' theory is supported, and that no additional experiments are required.

I listed several points to improve the manuscript. If the authors adequately addressed all of these concerns, I recommend publication in *Nature Communications*.

Major points

Based on the previous reports of Hohn's group (Chen et al. 2000; AEM 66;2062-2065) and Beremand's group (Tag et al. 2001; ref. 24), authors of ref. 5 proposed an alternative interpretation of the effects of disrupting *TRI6* on *TRI10* gene expressions. The idea that structural perturbation of the trichothecene cluster core region leads to deregulation of *TRI* gene expressions was supported by transcriptional analyses of *TRI10* in two distinct types of *TRI6* disruption mutants (Fig. 2E of ref. 5; with time-dependent expression profile of *TRI10*). The authors' conclusion of "**Repression of *TRI10* expression by *TRI6* and vice versa**" (line 146) contrasts sharply with the conclusion drawn from ref. 5 at this point. In addition, negative regulation of *TRI6* by *TRI10* (lines 151-160, **Fig. 1c**, **Fig. 2b**, and lines 301-303 of this manuscript) needs to be carefully mentioned, as forced expression of *hph* marker gene at the *TRI6* native locus may affect loosening the tightly packed heterochromatin structure of the trichothecene cluster core region (which seems to become more euchromatin-like by the action of Fgp1). Even if the authors do not agree with the theory, they should at least contrast their results with those described in ref. 5 and discuss the results in this manuscript. Specifically, please consider the following points:

(1) As to the *tri6* and *tri10* mutants, more descriptions of the gene cluster structure (ref. 14; co-authored by Jin-Rong Xu, one of the corresponding authors of this manuscript) are necessary for general readers. Please refer to the changes in the length of the cluster core region by replacing *TRI6* and *TRI10* with the *hph* cassette. Inferring from the available information described in ref. 14 (**Figure S1**; based on **Experimental procedures** for gene disruption, **Table S5** primer sequences used for the gene replacement, *hph* cassette described by Zhao *et al.*, 2004, and the PH-1 genome sequence database) the gene cluster core region seems to be extended to a small extent by the replacement in the *tri6* mutant, while that of the *tri10* is almost the same length. Is this interpretation correct?

(2) I am concerned about the possibility of observing artifacts caused by the *hph* cassette insertion, not the direct effects of *TRI6* or *TRI10* inactivation. Please mention about the direction of the *hph* cassette (and a promoter used for expression) in the *tri6* and *tri10* mutants. For example, transcription of the adjacent *TRI5* gene may be influenced in the *tri6* mutant, if the drug-resistant *hph* marker is actively transcribed (from a promoter subject to a regulation distinct from that of *TRI6*) from the original *TRI6* locus.

- Is the possibility of causing unusual *TRI5* and *TRI6* transcriptions by the genetic manipulations negligible?
- Are such influences, if any, affected by the transcription direction of *hph* in the marker cassette?

These points are important in interpreting the RNA-seq and qRT-PCR data of the mutants. Please discuss your thought on these points. It is helpful for readers if the illustrations of the *TRI6-TRI5-TRI10* loci of the wild type, *tri6* mutant, and *tri10* mutant are added to **Fig. 1**.

The most interesting results of this paper is the complete abrogation of toxin synthesis by mutating a Nasmith sequence (from GTGA to TGAG) upstream of *TRI6* (**Fig. 8**). Based on the result, the authors proposed a new model of transcriptional regulation of *TRI5* by RNA5P, Nasmith sequence, and Tri6p. As the model has a great impact on studies of trichothecene biosynthesis regulation, carefully describe the experimental results in more detail.

(3) Please describe the PCR verification data of the TM6 transformant that arose by double cross-over homologous recombination (**Fig. 8a**). **Fig. 8bc** contains the most important result. Negative selection with *tk* is more troublesome than positive selection with *hph*. Often, *tk* marker is mutated during negative selection, and also, unexpected illegitimate recombination tends to occur. For solidness of this paper, I suggest that the PCR verification data of strain TM6 is necessary in supplementary materials. On the contrary, the PCR verification data of very easy

positive selection to obtain ectopic transformants (**Fig. 6a** lower panel LT7 and LT10) are not necessary as displayed items in the main text.

(4) Transcriptions of anti-*TRI5* and RNA5P were low in the wild-type PH-1 strain (**Fig. 1c**), but this manuscript does not provide information of how low these transcript levels actually were. Their levels were set to 1.0 throughout this manuscript and the corresponding transcripts of transgenic strains were compared in qRT-PCR analyses. If they are compared to some appropriate internal control transcripts, such as those of *ubc* or *tub* genes, we can see the transcript level, which is very important.

In another word, how low were the anti-*TRI5* and RNA5P transcript levels compared to those of *TRI5* in PH-1? For example, if the anti-*TRI5* transcript level is 0.00000001 % of the *TRI5* mRNA in PH-1, does it make sense to compare their expression levels between PH-1 and TM6 strains (**Fig. 8c**)? The same notion also applies to **Fig. 2**, **Fig. 3b**, **Fig. 4c**, **Fig. 5c**, **Fig. 6c**, **Fig. 7c**, and **Supplementary Fig. 3c**.

Therefore, please describe the transcript levels of RNA5P, *TRI5*, anti-*TRI5*, *TRI6*, and anti-*TRI6* in PH-1 relative to that of an appropriate control housekeeping gene.

Other points

(5) Lines 173-175:

Why was the *TRI6* transcript decreased by the replacement of the “185-bp region” with the *hph* cassette? How about discussing the possibility of increased *TRI6* mRNA instability due to the elimination of the immediate downstream terminator region? The authors only mentioned about reduced transcription of sense *TRI6* mRNA.

(6) Lines 211-213:

The authors replaced the 834-bp fragment upstream from *TRI5* ORF with an 834-bp *TRI12* promoter fragment. General readers feel why both fragments are exactly the same length. Was it only by chance or intended to be the same length? If this experimental design was based on the idea described in ref. 5, authors should appropriately explain the reason citing the literature.

(7) **Supplementary Fig. 3** legend:

The experimental conditions are described quite ambiguously and it is difficult to follow which region (in panel **a**) corresponds to N, C, and T (in panel **b**). Please clearly indicate the names of “specific primers“, which may be listed in **Supplementary Table 3**, and show the primer

annealing site in panel **a**. In panel **b**, how was the cDNA synthesized? Unambiguously describe the culture condition (LTB?) and period (3 days?).

(8) Lines 261-264:

Is the Nasmith sequence present at this position in other *Fusarium* strains? (I am skeptical of Nasmith sequence)

Minor points

(9) Line 49:

Why is ref. 5 placed here? This reference has a different meaning and it is not appropriate (or distorted) to be cited at this paragraph.

(10) Lines 56-68:

As to the second paragraph describing the history of studies on the trichothecene gene cluster, consider to choose appropriate citations.

(11) Line 152 and lines 155-156:

Fig. 2a should be **Fig. 2b**?

(12) Lines 215-216 (related to **Fig. 5b** experimental scheme and missing PCR verification data):

Please include the evidence that the designed double cross-over homologous recombination occurred by negative selection in supplementary materials, by showing the PCR verification data of P_{TR112}-*TRI5* transformant.

(13) Lines 246-247 (related to **Fig. 7a** experimental scheme and missing PCR verification data):

Same as above.

(14) Lines 451-454:

Primer pairs used for identification of the seven transcripts are not listed; that is, only 6 primer pairs are shown and one primer pair is missing. Information of primers used for qRT-PCR should be added in **supplementary Table 3** (*e.g.*, clearly indicate target transcripts, such as *TRI5* or antisense-*TRI5*, in addition to the primer names and sequences). Primer TR2 is depicted in **Fig. 3a** and **Fig. 4a**, but a primer with such a name is missing in line 435 and **supplementary Table 3**. Also depict the primer annealing site in **Fig. 2c**. Perhaps primer pair for qRT-PCR of

TRI6 or antisense-*TRI6* is missing in line 453. With such ambiguous descriptions, I could not check the experimental design.

(15) Lines 206-208 and **Fig. 4d** legend:

The authors should mention more about the positive control (what is FgESA1?) used for the WB analysis. General readers may be curious about the reason why histone deacetylase (FgESA1) has been chosen as the positive control in this experiment. Basically, the presence or absence of a certain protein should carefully be considered. For example, my group fails to detect the Tri10p protein from a wild-type *F. graminearum* strain under toxin-inducing conditions, but the failure to detect a Tri10p band in WB analysis does not imply its absence when trichothecene pathway genes are actively transcribed. I speculate that the authors' logic of showing HAT as the positive control here is to demonstrate the high sensitivity of their experiment.

“When toxin synthesis is activated, the amount of HAT is known to be low (add ref.) but even such a limited amount of protein could successfully be detected by anti-FLAG antibody, while a hypothetical RNA5P protein could not be detected on the same WB. Thus, RNA5P appears not to encode a protein.”

If so, please clarify the point.

(16) Line 199 and **Fig. 4**:

Please clarify the following points:

“TF2” should be “TR2” in **Fig. 4a**? But “TR2” is missing in **Supplementary Table 3** (I could only find primer name “TR2R”).

“amplified” should be “amplify”?

Speculating from the information in **Fig. 4b**, perhaps the sentence may be corrected to “Whereas primer pairs “5PRT1/5PRT2” and “TF1/**TR2**” could amplify the **0.1-kb and 0.12-kb bands, respectively**, no band could be amplified with primer pairs “5PRT12 and “**TR2**” (Fig. 4b)”? Please explain whether the main text sentence is correct or **Fig. 4a** is correct. The manuscript needs to be thoroughly checked.

(17) **Fig. 2c**:

What are “T6RTF” and “T6RTR”?; were they used for qRT-PCR? They are missing M & M section and **Supplementary Table 3**. I could only find similar but distinct primer names “T6F” and “T6R” (line 453), and “T6RTNF” and “T6RTNR” in **Supplementary Table 3**. The manuscript should be carefully checked before publication.

(18) **Fig. 3a:**

Same as above. Primers “TR2” and “AR4” are missing in M & M section (lines 451-454) and **Supplementary Table 3**. For other Figures with primers, please thoroughly the primer names for accuracy.

Reviewer #2 (Remarks to the Author):

The manuscript presents a comprehensive analysis of the expression of regulatory genes for DON biosynthesis. The stranded RNA-seq approach used in the analysis allows the regulation process to be understood in an accurate manner. In addition, all the strains used in this work unravel the regulation process clearly. The paper is very well written and the figures help to understand the complexity of the results. The methods are well presented, but in the description of the research material I miss information on the number of biological replicates used for the differential expression analysis. It would be interesting to study the behaviour of the lncRNA found in this analysis during wheat infection. From my point of view, this work can be published in its current form. Only a few minor issues should be addressed:

Line 52:

"The TRI5 trichodiene synthase gene is the first virulence factor.."

The gene that encodes TRI5 trichodiene synthase is the first virulence factor..

Line 56:

First time you mention this species: *Fusarium sporotrichioides*

Line 354:

Supplementary Table 1 doesn't correspond to the strains used in this study.

Line 433

Please, specify the number of biological replicates used for the DE analysis.

Line 448:

Please, explain what is the reason why the strains were isolated during 5 days for the RNA-seq and 3 days for the qPCR assay.

Line 455:

There is a typo.

Line 292:

TRI14 encodes a protein that is specifically required for infection because it is alternatively spliced? Explain please.

Supplementary Fig. 1:

It is missing the "c".

Reviewer #3 (Remarks to the Author):

Although function of lcrRNA is myriad, nomenclature and identification can be clearly defined based on the consensus statement <https://doi.org/10.1038/s41580-022-00566-8>. Authors have provided evidence to suggest a role of lcrRNA in the regulation of TRI5, however, the manuscript could be aided with more information (detailed below).

Introduction, results and the Figures are well written. Discussion section could definitely use some editing and more focus.

Overall, the study is worth publishing as it sheds novel insights into the regulation of secondary metabolism in fungi.

The authors Huang et al., aim to show that TRI5 in the phytopathogen can be regulated in multiple ways. TRI5 is the first enzyme involved in the biosynthesis of DON in the phytopathogen *Fusarium graminearum*.

1. The use of RNAseq and RT-qPCR, the authors demonstrate that TRI5 is regulated by TRI6 and TRI10 through suppression of antisense TRI5 transcripts.

Q: Will constitutive expression of either TRI6 in tri0 mutant or constitutive expression TRI10 in the tri6 mutant alleviate the repression – do you need both 6 and 10?

Line 187: no evidence is provided to say "degradation of sense transcripts of TRI5..."

2. The RNAseq data also suggested that lcrRNAs are involved in the regulation of TRI5.

Q: Did comparison to other RNAseq database reveal potential CDS/small ORFs in the intergenic region between TRI6 and TRI5?

- Please indicate the version of PH1 reference genome used.

- Figure 1: please indicate transcript sizes and the distance between the genes in the cluster (specifically TRI5 and 6).

Q: How was TSS determined for the RNA5P in Figure 4.

- provide the sequence of the lcrRNA (start and end) and how did they decide which frame to use for tagging the RNA5P?

- To suggest that "RNA5P does not encode proteins" (line 208) is premature (taken previous point into account). The authors could also use in vitro translation.

The authors replaced the TRI5 promoter with TRI12 promoter

Q: did RNA seq data reveal any potential transcripts from the TRI12 promoter region in the WT strain?

In situ overexpression by TrpC promoter.

Please ensure the transformants TR1, TR2, ATR1, ATR2 in Figure 7 is referenced in the text.

Discussion:

Could use more editing and focus. Please rephrase:

Line 283: "global regulator" is not defined by the quantity of genes, but by the number of pathways it influences.

Line 289: both TRI6 and TRI10 are necessary for DON regulation. Its presumptuous to say one is more important than the other.

Line 297: clarify " qRT-PCR assays that do not distinguish sense and antisense....."

Line 308- check Shostak et al., <https://doi.org/10.1111/mmi.14575>

Line 316: no evidence for this

Line 326: use more updated references, specifically the consensus statement regarding nomenclature and function of lcrRNAs <https://doi.org/10.1038/s41580-022-00566-8>. doi: 10.3389/fmicb.2021.638617

Line 337: DON is made by specific species of fungi and the statement in line 337 is a "truism"

Line 338: secondary mechanism or secondary metabolism?

Line 338-339: rephrase. Consider "In yeasts, plants, and animals cis-acting some lcrRNAs mediate.....(PRC2), while others repress....."

Other issues:

Iine 148: "particularly the full-length TRI10 transcripts (Fig 1c)" – NOT shown

Line 168: typo: coving to covering

Figure 4c: what is L3F?

Figure 4: legend. What is FgESA1-3 x FLAG?

Figure 6a. primer pair combination on top of the gel should read P27F/5PR

Below are our point-by-point responses to reviewers' comments.

Reviewer #1 (Remarks to the Author):

This is an interesting paper introducing a new model of trichothecene biosynthesis regulation by Tri6p. By mutating the 4-bp Nasmith sequence (GTGA repeats, separated by 8 bp or less) at its native locus, the authors showed positive regulation of TRI5 through repression of the RNA5P transcript. Although my group think of a quite different model of regulatory mechanism of toxin production, the authors' model (Fig. 8d: which assumes suppression of RNA5P transcription by binding of Tri6p to the Nasmith sequence) has a large impact and could stimulate research progress and discussion in the relevant community. Although other reviewers may request EMSA data as Nasmith et al. (ref. 22) did so, I do not think them necessary in this paper. Indeed, full-length Tri6-HA recombinant protein cannot be easily obtained in an active form, as described in ref. 22 (page 10, left column last paragraph of ref. 22), and we had experienced difficulties in obtaining active form of recombinant Tri6p in *E. coli*. Provided that the molecular genetic data on the role of the Nasmith sequence are satisfactorily presented in this manuscript, I think the authors' theory is supported, and that no additional experiments are required. I listed several points to improve the manuscript. If the authors adequately addressed all of these concerns, I recommend publication in Nature Communications.

Response: Thanks for the comments.

Major point.

Based on the previous reports of Hohn's group (Chen et al. 2000; AEM 66;2062-2065) and Beremand's group (Tag et al. 2001; ref. 24), authors of ref. 5 proposed an alternative interpretation of the effects of disrupting TRI6 on TRI10 gene expressions. The idea that structural perturbation of the trichothecene cluster core region leads to deregulation of TRI gene expressions was supported by transcriptional analyses of TRI10 in two distinct types of TRI6 disruption mutants (Fig. 2E of ref. 5; with time-dependent expression profile of TRI10). The authors' conclusion of "Repression of TRI10 expression by TRI6 and vice versa" (line 146) contrasts sharply with the conclusion drawn from ref. 5 at this point. In addition, negative regulation of TRI6 by TRI10 (lines 151-160, Fig. 1c, Fig. 2b, and lines 301-303 of this manuscript) needs to be carefully mentioned, as forced expression of hph marker gene at the TRI6 native locus may affect loosening the tightly packed heterochromatin structure of the trichothecene cluster core region (which seems to become more euchromatin-like by the action of Fgp1). Even if the authors do not agree with the theory, they should at least contrast their results with those described in ref. 5 and discuss the results in this manuscript.

Response: Thanks for the comments. As described in ref. 5 (ref. 14 in the revised manuscript, Liew et al., *Front. Microbiol.*, 2023), the expression of *TRI10* was elevated in the *Tri6 tk* mutant but not in the *Tri6-nsm* mutant, suggesting the structural extension of the trichothecene cluster core region resulted in a transcriptional change of *TRI* genes.

However, unlike the *Tri6 tk* strain (Liew et al., *Front. Microbiol.*, 2023) that generated a 7.0-kb extension in the *TRI* gene cluster, the gene replacement event in the *tri6* mutant (Seong et al., *Mol. Microbiol.*, 2009) resulted only 248 -bp elongation at the *TRI6* locus (which appeared to have less impact on chromosome structure), which is 2.6-kp upstream from *TRI5* (Fig. 1a). Furthermore, the defect of the *tri6* mutant in DON biosynthesis was complemented by ectopic expression of *TRI6* (Seong et al., *Mol. Microbiol.*, 2009), confirming that the phenotype

observed in the mutant is directly related to *TRI6* deletion. Similarly, the *tri10* deletion mutant was complemented by ectopic expression of *TRI10* and the gene replacement event at this locus resulted in a 575 -bp shortening (Seong et al., *Mol. Microbiol.*, 2009). In addition, changes in chromatin structures usually create an environment that is permissive for similar activation or repression of neighboring genes (Sproul et al., *Nat. Rev. Genet.*, 2005), however, the *tri6* deletion mutant was significantly reduced in *TRI5* expression but increased in *TRI10* expression although all three genes in the *TRI6-TRI5-TRI10* region are transcribed in the same direction (Fig. 1a). The opposite effects of *TRI6* deletion on the expression of *TRI5* and *TRI10* (Fig. 1c; Seong et al., *Mol. Microbiol.*, 2009) further indicate the transcriptional regulation on *TRI* genes by *TRI6*.

Due to the difference in culture conditions (YS_60 vs LTB) and wild-type strains (JCM 9873 vs PH-1) used by Liew and colleagues (Liew et al., *Front. Microbiol.*, 2023) and in our studies, to experimentally address this comment, we generated a Tri6-nsm strain in the wild-type strain PH-1 by inserting the stop codon at the middle region (The C2H2 zinc finger DNA binding domain is located at the C-terminal of Tri6, which was disrupted by this insertion) of *TRI6* (⁴⁸⁴TAA⁴⁸⁶) by the knock-out and knock-in approach (See the figure below). Because DON is a secondary metabolite, we normally assay DON production and *TRI* gene expression in cultures of strain PH-1 that have been incubated for at least for 72 h. When cultured in LTB medium for 72h, the expression of *TRI10* was increased in the Tri6-nsm strain in comparison with the wild type (See the figure below), confirming the repression of *TRI10* expression by *TRI6*. In contrast, Liew and colleagues (Liew et al., *Front. Microbiol.*, 2023) found the expression of *TRI10* only slightly up-regulated (1.34-fold) in the Tri6-nsm strain when cultured for 62 h (not increased in 36 h and 48 h samples). This difference could be due to the time point for the qPCR assay (72 h vs 62 h). It is also noteworthy that the nonsense mutation TAG was introduced to the fourth residue M (A¹⁰TG¹² to TAG) in JCM 9873 (Liew et al., *Front. Microbiol.*, 2023). Translation from downstream in-frame ATG codons or CTG/GTG codons (Wei et al., *J. Biol. Chem.*, 2013) may lead to the production of truncated Tri6 proteins (with a truncation at the N terminal of the Tri6 protein, a downstream in-frame ATG codon or some non-ATG codons of *TRI6* might be also functional for translation initiation). In fact, recently we have observed such phenomenon (truncation of N-terminal region due to site-specific mutations) with *FgTAD2*, an essential gene in *F. graminearum*.

Overall, based on the experimental data and a review of previous publications, *TRI6* is involved in the transcriptional regulation of *TRI10* at least when cultured in LTB medium for 72 h. Given that Liew and colleagues observed an interesting phenomenon and raised an important point on mutational effects of gene disruption on chromatin structural changes, we have revised the discussion to include this important perspective.

Generation of the *TRI6-nsm* strain using a two-step transformation process.

a Schematic drawing of the primers used to generate the *TRI6-nsm* strain by a two-step transformation process. For the first step, the upstream and downstream flanking sequences of the *TRI6* gene were amplified with primer pairs T6HT-1F/T6HT-2R and T6HT-3F/T6HT-4R and connected to the *hph-tk* cassette to generate the *tri6 tk* mutant. For the second step, a dysfunctional copy of the *TRI6* gene containing a nonsense mutation was generated for the replacement of the *hph-tk* cassette. **b** PCR verification of the *tri6 tk* mutant and *TRI6-nsm* strain using primers specific to *TRI6* (AT6-5F and AT6-6R). The expected sizes of fragments were labeled on the left of the gel images. **c** Assayed for the *TRI10* expression by qRT-PCR with RNA isolated from 3-day-old LT6 cultures of the wild-type PH-1 (arbitrarily set to 1) and the *TRI6-nsm* strain. Mean and standard deviation were estimated with data from three independent replicates ($n = 3$).

Specifically, please consider the following points:

(1) As to the *tri6* and *tri10* mutants, more descriptions of the gene cluster structure (ref. 14; co-authored by Jin-Rong Xu, one of the corresponding authors of this manuscript) are necessary for general readers. Please refer to the changes in the length of the cluster core region by replacing *TRI6* and *TRI10* with the *hph* cassette. Inferring from the available information described in ref. 14 (Figure S1; based on Experimental procedures for gene disruption, Table S5 primer sequences used for the gene replacement, *hph* cassette described by Zhao et al., 2004, and the PH-1 genome sequence database) the gene cluster core region seems to be extended to a small extent by the replacement in the *tri6* mutant, while that of the *tri10* is almost the same length. Is this interpretation correct?

Response: As suggested, we added more detailed descriptions of the core *TRI* gene cluster (A detailed description of *TRI* genes can be found in the introduction section). We also revised Fig. 1a to provide detailed information on the changes in the size of the core *TRI* gene cluster in the *tri6* and *tri10* deletion mutants. Whereas the *TRI5-TRI6* region was 248 -bp longer in the *tri6* mutant, the *TRI5-TRI10* region was 575 -bp shorter in the *tri10* mutant in comparison with the wild type (Fig. 1a) due to the gene replacement events. The gene replacement of *TRI6* and *TRI10* with a gene conferring resistance to hygromycin B (*hph*) was shown in the Supplementary Fig. 1 of the previous publication (Seong et al., *Mol. Microbiol.*, 2009).

(2) I am concerned about the possibility of observing artifacts caused by the *hph* cassette insertion, not the direct effects of *TRI6* or *TRI10* inactivation. Please mention about the direction of the *hph* cassette (and a promoter used for expression) in the *tri6* and *tri10* mutants. For example, transcription of the adjacent *TRI5* gene may be influenced in the *tri6* mutant, if the drug-resistant *hph* marker is actively transcribed (from a promoter subject to a regulation distinct from that of *TRI6*) from the original *TRI6* locus.

- Is the possibility of causing unusual *TRI5* and *TRI6* transcriptions by the genetic manipulations negligible?

- Are such influences, if any, affected by the transcription direction of *hph* in the marker cassette?

Response: In the revised manuscript, we added Fig. 1a to show the directions of the *hph* cassette and changes in the size of the *TRI6-TRI5-TRI10* region in the *tri6* and *tri10* mutants. As shown in Fig. 1a, *TRI6*, *TRI5*, *TRI10*, and the *hph* cassette have the same transcription direction. As we explained above, deletion of *TRI6* reduced the expression of *TRI5* but increased *TRI10*

expression. Deletion of *TRI10* also reduced the expression of *TRI5* but increased *TRI6* expression. Relative to *TRI5*, the transcription direction of the *hph* cassette was in the same direction in the *tri6* mutant but in the opposite direction in the *tri10* mutant because *TRI5* is between *TRI6* and *TRI10*. Therefore, it is unlikely that changes in the expression of *TRI5* and *TRI10* are due to the transcription of the *hph* cassette.

Nevertheless, to address this comment experimentally, we generated the *tri6* and *tri10* deletion mutants with the *hph* cassette in the opposite direction. In these new *tri6* and *tri10* mutants, *TRI5* expression and DON production were also significantly reduced (Supplementary Fig. 2). These results indicated that the transcription direction of *hph* is not an important factor for reduced *TRI5* expression in the *tri6* or *tri10* mutant. Therefore, changes in the *TRI5* expression in the *tri6* or *tri10* mutant are not related to the direction of the *hph* cassette.

The most interesting results of this paper is the complete abrogation of toxin synthesis by mutating a single Nasmith sequence (from GTGA to TGAG) upstream of *TRI5* (Fig. 8). Based on the result, the authors proposed a new model of transcriptional regulation of *TRI5* by RNA5P, Nasmith sequence, and Tri6p. As the model has a great impact on studies of trichothecene biosynthesis regulation, carefully describe the experimental results in more detail.

Response: As suggested, more details were added to the Results related to the mutants with the GTGA-to-TGAG mutation. During revision, we isolated an additional transformant with the GTGA-to-TGAG mutation and assayed the expression of RNA5P and *TRI5* as well as DON production. Similar to the original transformant TM6, the new transformant TM66 was also increased in RNA5P expression, and significantly reduced in *TRI5* expression and DON biosynthesis. These results confirmed our observation on the regulatory role of RNA5P and the importance of this GTGA site (Data related to TM66 were added to the revised Fig. 8).

(3) Please describe the PCR verification data of the TM6 transformant that arose by double cross-over homologous recombination (Fig. 8a). Fig. 8bc contains the most important result. Negative selection with *tk* is more troublesome than positive selection with *hph*. Often, *tk* marker is mutated during negative selection, and also, unexpected illegitimate recombination tends to occur. For solidness of this paper, I suggest that the PCR verification data of strain TM6 is necessary in supplementary materials. On the contrary, the PCR verification data of very easy positive selection to obtain ectopic transformants (Fig. 6a lower panel LT7 and LT10) are not necessary as displayed items in the main text.

Response: As suggested, we add Fig. 8b to show the PCR verification and sequencing results on transformants with the expected GTGA-to-TGAG mutation. Primer pairs T5P-F and T5Y-8R were used to amplify and sequence the promoter region with this GTGA region were also added to Fig. 8b.

(4) Transcriptions of anti-*TRI5* and RNA5P were low in the wild-type PH-1 strain (Fig. 1c), but this manuscript does not provide information of how low these transcript levels actually were. Their levels were set to 1.0 throughout this manuscript and the corresponding transcripts of transgenic strains were compared in qRT-PCR analyses. If they are compared to some appropriate internal control transcripts, such as those of *ubc* or *tub* genes, we can see the transcript level, which is very important.

In another word, how low were the anti-*TRI5* and RNA5P transcript levels compared to those of *TRI5* in PH-1? For example, if the anti-*TRI5* transcript level is 0.00000001 % of the

TRI5 mRNA in PH-1, does it make sense to compare their expression levels between PH-1 and TM6 strains (Fig. 8c)? The same notion also applies to Fig. 2, Fig. 3b, Fig. 4c, Fig. 5c, Fig. 6c, Fig. 7c, and Supplementary Fig. 3c.

Therefore, please describe the transcript levels of RNA5P, TRI5, anti-TRI5, TRI6, and antiTRI6 in PH-1 relative to that of an appropriate control housekeeping gene.

Response: As suggested, we counted the TPM (transcripts per million) values of RNA5P, *TRI5*, anti-*TRI5*, *TRI6*, anti*TRI6*, and *ACTIN* (a housekeeping gene that was used as the internal control in qPCR assay) in RNA-seq data of the wild-type strain PH-1. The expression level of RNA5P and anti-TRI5 was less than 2% of that of *ACTIN*. Related data were presented in Supplementary Table 5.

(5) Lines 173-175: Why was the TRI6 transcript decreased by the replacement of the “185-bp region” with the hph cassette? How about discussing the possibility of increased TRI6 mRNA instability due to the elimination of the immediate downstream terminator region? The authors only mentioned about reduced transcription of sense TRI6 mRNA.

Response: As suggested, we added a related discussion in the revised manuscript. Because this 185-bp region is behind the 3'-UTR of *TRI6*, it may not affect the stability of sense transcripts of *TRI6*. Based on the transcription stop site of *TRI6* sense transcripts (determined by RNA-seq data, Fig. 1d), this 185-bp region likely contains the terminator sequence of *TRI6*. Therefore, most likely, deletion of this 185-bp region may affect the efficiency of *TRI6* transcription.

(6) Lines 211-213: The authors replaced the 834-bp fragment upstream from TRI5 ORF with an 834-bp TRI12 promoter fragment. General readers feel why both fragments are exactly the same length. Was it only by chance or intended to be the same length? If this experimental design was based on the idea described in ref. 5, authors should appropriately explain the reason citing the literature.

Response: Indeed, we used the same length fragments for promoter swapping because of ref. 5 (ref. 14 in the revised manuscript, Liew et al., *Front. Microbiol.*, 2023) and concerns about size differences in the chromatin structure of the *TRI6-TRI5-TRI10* region. In the revised manuscript, we added the following sentence to provide an explanation. “The size of the promoter regions used for promoter swapping was kept similar to avoid possible effects of size changes on local chromatin structures in the *TRI* gene cluster (Liew et al., *Front. Microbiol.*, 2023)”.

(7) Supplementary Fig. 3 legend: The experimental conditions are described quite ambiguously and it is difficult to follow which region (in panel a) corresponds to N, C, and T (in panel b). Please clearly indicate the names of “specific primers”, which may be listed in Supplementary Table 3, and show the primer annealing site in panel a. In panel b, how was the cDNA synthesized? Unambiguously describe the culture condition (LTB?) and period (3 days?).

Response: As suggested, we revised the figure and figure legend to include specific information of primers and culture conditions.

(8) Lines 261-264: Is the Nasmith sequence present at this position in other *Fusarium* strains? (I am skeptical of Nasmith sequence)

Response: The putative Tri6-binding site [GTGA/TCAC-N(0-8)-GTGA/TCAC] is present in the same position (-1411 from the transcription start site) of the upstream region of *TRI5* in both *F. graminearum* and *F. culmorum* but not in *F. sporotrichioides*. However, in *F. sporotrichioides*,

the GTGAAACATGTCAC sequence that meets the GTGA-N6-TCAC consensus was identified at the other position (-1350 from the transcription start site). These results suggested that different *Fusarium* species may vary in the Tri6-binding sequences or the position of Tri6-binding sites upstream of *TRI5*.

Minor point

(9) Line 49: Why is ref. 5 placed here? This reference has a different meaning and it is not appropriate (or distorted) to be cited at this paragraph.

Response: This reference has been replaced with another review paper.

(10) Lines 56-68: As to the second paragraph describing the history of studies on the trichothecene gene cluster, consider to choose appropriate citations.

Response: The citations were updated during revision.

(11) Line 152 and lines 155-156: Fig. 2a should be Fig. 2b?

Response: Corrected.

(12) Lines 215-216 (related to Fig. 5b experimental scheme and missing PCR verification data): Please include the evidence that the designed double cross-over homologous recombination occurred by negative selection in supplementary materials, by showing the PCR verification data of PTRI12-TRI5 transformant.

Response: As suggested, PCR verification data of the P_{TRI12}-TRI5 transformant were provided in Supplementary Fig. 7.

(13) Lines 246-247 (related to Fig. 7a experimental scheme and missing PCR verification data): Same as above.

Response: As suggested, PCR verification data were added in revised Fig. 7a.

(14) Lines 451-454: Primer pairs used for identification of the seven transcripts are not listed; that is, only 6 primer pairs are shown and one primer pair is missing. Information of primers used for qRT-PCR should be added in supplementary Table 3 (e.g., clearly indicate target transcripts, such as TRI5 or antisense-TRI5, in addition to the primer names and sequences). Primer TR2 is depicted in Fig. 3a and Fig. 4a, but a primer with such a name is missing in line 435 and supplementary Table 3. Also depict the primer annealing site in Fig. 2c. Perhaps primer pair for qRT-PCR of TRI6 or antisense-TRI6 is missing in line 453. With such ambiguous descriptions, I could not check the experimental design.

Response: The supplementary table for primers (Supplementary Table 6 in the revised manuscript) was revised to categorize all the primers based on their applications. A column was added to describe the targets of these primers used for qRT-PCR assay. We also double-checked to make sure all the primers used in this study were listed in Supplementary Table 6.

(15) Lines 206-208 and Fig. 4d legend: The authors should mention more about the positive control (what is FgESA1?) used for the WB analysis. General readers may be curious about the reason why histone acetylase (FgESA1) has been chosen as the positive control in this experiment. Basically, the presence or absence of a certain protein should carefully be

considered. For example, my group fails to detect the Tri10p protein from a wild-type *F. graminearum* strain under toxin-inducing conditions, but the failure to detect a Tri10p band in WB analysis does not imply its absence when trichothecene pathway genes are actively transcribed. I speculate that the authors' logic of showing HAT as the positive control here is to demonstrate the high sensitivity of their experiment.

“When toxin synthesis is activated, the amount of HAT is known to be low (add ref.) but even such a limited amount of protein could successfully be detected by anti-FLAG antibody, while a hypothetical RNA5P protein could not be detected on the same WB. Thus, RNA5P appears not to encode a protein.” If so, please clarify the point.

Response: Thanks for the comment. Even under the control of this high-efficiency RP27 promoter, the anti-FLAG antibody failed to detect any band on western blots of total proteins isolated from the resulting RNA5P-3xFLAG transformants cultured in LTB (Fig. 4d). In contrast, the histone acetyltransferase encoded by *FgESAI* which has a low expression during the DON-producing stage (Jiang et al., *PLoS Genet.*, 2020), is successfully detected by anti-FLAG antibody in the same western blot analysis”. This point has been clarified in the revised manuscript. The following sentence was added to the revised manuscript. “As the control, the expression of *FgEsa1* histone acetyltransferase was detected in the *FgESAI*-3xFLAG transformant (Jiang et al., *PLoS Genet.*, 2020) under the same conditions (Fig. 4d)”.

(16) Line 199 and Fig. 4: Please clarify the following points: “TF2” should be “TR2” in Fig. 4a? But “TR2” is missing in Supplementary Table 3 (I could only find primer name “TR2R”). “amplified” should be “amplify”? Speculating from the information in Fig. 4b, perhaps the sentence may be corrected to “Whereas primer pairs “5PRT1/5PRT2” and “TF1/TR2” could amplify the 0.1-kb and 0.12-kb bands, respectively, no band could be amplified with primer pairs “5PRT12 and “TR2” (Fig. 4b)? Please explain whether the main text sentence is correct or Fig. 4a is correct. The manuscript needs to be thoroughly checked.

Response: During revision, we thoroughly checked the entire manuscript and made necessary revisions in the main text, related figure legend, and Supplementary Table 6 (with the primer information).

(17) Fig. 2c: What are “T6RTF” and “T6RTR”?; were they used for qRT-PCR? They are missing M & M section and Supplementary Table 3. I could only find similar but distinct primer names “T6F” and “T6R” (line 453), and “T6RTNF” and “T6RTNR” in Supplementary Table 3. The manuscript should be carefully checked before publication.

Response: The primer pair T6RTF/T6RTR was used for qRT-PCR assays of *TRI6* and anti-*TRI6*. The related figure legend and Supplementary Table 6 (with the information of primers) were revised during revision.

(18) Fig. 3a: Same as above. Primers “TR2” and “AR4” are missing in M & M section (lines 451-454) and Supplementary Table 3. For other Figures with primers, please thoroughly check the primer names for accuracy.

Response: As suggested, we thoroughly checked our manuscript and made the necessary revisions in the main text, related figure legend, and Supplementary Table 6. The revised Supplementary Table 6 lists all the primers used in this study and has a column listing their applications.

Reviewer #2 (Remarks to the Author):

The manuscript presents a comprehensive analysis of the expression of regulatory genes for DON biosynthesis. The stranded RNA-seq approach used in the analysis allows the regulation process to be understood in an accurate manner. In addition, all the strains used in this work unravel the regulation process clearly. The paper is very well written and the figures help to understand the complexity of the results. The methods are well presented, but in the description of the research material I miss information on the number of biological replicates used for the differential expression analysis. It would be interesting to study the behaviour of the lncRNA found in this analysis during wheat infection. From my point of view, this work can be published in its current form. Only a few minor issues should be addressed:

Response: Thanks for the comments. As suggested, we added information on the number of biological replicates in the revised manuscript.

Line 52:

“The TRI5 trichodiene synthase gene is the first virulence factor...”

The gene that encodes TRI5 trichodiene synthase is the first virulence factor...

Response: Revised as suggested.

Line 56:

First time you mention this species: *Fusarium sporotrichioides*

Response: Revised as suggested.

Line 354:

Supplementary Table 1 doesn't correspond to the strains used in this study.

Response: Corrected.

Line 433

Please, specify the number of biological replicates used for the DE analysis.

Response: Related information was added as suggested.

Line 448:

Please, explain what is the reason why the strains were isolated during 5 days for the RNA-seq and 3 days for the qPCR assay.

Response: Related descriptions were revised to clarify confusion. Total RNAs were isolated from hyphae harvested from 3-day-old LTB cultures for both qPCR and RNA-seq. Conidia used to inoculate LTB were collected from 5-day-old CMC cultures.

Line 455:

There is a typo.

Response: Corrected.

Line 292:

TRI14 encodes a protein that is specifically required for infection because it is alternatively spliced? Explain please.

Response: This sentence was revised to provide explanations and avoid confusion. *TRI14*, a *TRI* gene with unknown function, appears to play different roles in DON production in axenic cultures and during plant infection. Whereas deletion of *TRI14* had no obvious effect on DON biosynthesis in cultures, the DON production in inoculated wheat kernels was significantly reduced. Similar to the *tri5* mutant, infection of the *tri14* mutant was restricted to the inoculated wheat kernels (Dyer et al., *J. Agric. Food Chem.*, 2005). In the revised manuscript, we added the following sentence to provide an explanation. “In *F. graminearum*, differences in DON biosynthesis between axenic cultures and infected plant tissues have observed in mutants deleted of *TRI14* that encodes a hypothetical protein of no know functions”.

Supplementary Fig. 1:
It is missing the “c”.

Response: Corrected.

Reviewer #3 (Remarks to the Author):

Although function of lcrRNA is myriad, nomenclature and identification can be clearly defined based on the consensus statement <https://doi.org/10.1038/s41580-022-00566-8>. Authors have provided evidence to suggest a role of lcrRNA in the regulation of *TRI5*, however, the manuscript could be aided with more information (detailed below).

Introduction, results and the Figures are well written. Discussion section could definitely use some editing and more focus.

Overall, the study is worth publishing as it sheds novel insights into the regulation of secondary metabolism in fungi.

Response: Thanks for the comments. The discussion has been revised to be clearer. The suggested reference (Mattick et al., *Nat. Rev. Mol. Cell Biol.*, 2023) was cited in the revised manuscript.

The authors Huang et al., aim to show that *TRI5* in the phytopathogen can be regulated in multiple ways. *TRI5* is the first enzyme involved in the biosynthesis of DON in the phytopathogen *Fusarium graminearum*.

1. The use of RNAseq and RT-qPCR, the authors demonstrate that *TRI5* is regulated by *TRI6* and *TRI10* through suppression of antisense *TRI5* transcripts.

Q: Will constitutive expression of either *TRI6* in *tri10* mutant or constitutive expression *TRI10* in the *tri6* mutant alleviate the repression – do you need both 6 and 10?

Response: Thanks for the comment. To further characterize the functional relationship between *TRI6* and *TRI10* in a separate study, we have generated the P_{RP27} -*TRI6* and P_{RP27} -*TRI10* overexpression constructs and transformed them into the wild type and the *tri10* or *tri6* mutant. Overexpression of *TRI10* resulted in increased *TRI5* expression and DON production in the wild type. However, in the *tri6* mutant, overexpression of *TRI10* had no effect on *TRI5* expression or DON production, indicating the requirement of *TRI6* for the effects of *TRI10* overexpression. Interestingly, subapical hyphal swelling was stimulated by *TRI10* overexpression in both the wild type or *tri6* deletion background, indicating that overexpression of *TRI10* has both *TRI6*-dependent and independent effects. For *TRI6*, its overexpression did not affect *TRI5* expression or DON production in the *tri10* mutant as well as the wild type. These results further indicate that both *TRI6* and *TRI10* are essential for *TRI* gene expression and DON biosynthesis although

TRI10 plays a more important regulatory role than *TRI6*. Overall, constitutive expression of *TRI10* in the *tri6* mutant or constitutive expression of *TRI6* in the *tri10* mutant failed to induce *TRI5* gene expression and DON biosynthesis. As suggested, we also assayed the expression level of antisense-*TRI5*. Constitutive expression of *TRI10* in the *tri6* mutant or constitutive expression of *TRI6* in the *tri10* mutant also failed to increase the expression level of antisense-*TRI5* (See the figure below).

Currently, we are studying the physical interactions between Tri6 and Tri10 proteins and the effects of their interactions on DNA binding (binding sites). We prefer to present results from these experiments on constitutive expression of *TRI10* in the *tri6* mutant or constitutive expression of *TRI6* in the *tri10* mutant in a separate manuscript on further characterization of the functional relationship between Tri6 and Tri10 (Manuscript preparation under the way).

The effects of *TRI10* and *TRI6* overexpression in the *tri6* and *tri10* deletion mutants.

a The expression of *TRI10* and *TRI5* assayed by qRT-PCR with RNA isolated from 3-day-old LTB cultures of the wild-type strain PH-1 (arbitrarily set to 1), P_{RP27} -*TRI10* transformant (*TRI10*-OE), *tri6* mutant, and *tri6* P_{RP27} -*TRI10* transformant (*TRI10*-OE/*tri6*). **b** DON production in 7-day-old LTB cultures of the labelled strains. **c** LTB cultures of the labelled strains were examined for bulbous structures (marked with arrows). Bar = 20 µm. **d** The expression of anti-*TRI5* assayed by qRT-PCR with RNA isolated from 3-day-old LTB cultures

of *tri6* mutant (arbitrarily set to 1), and *tri6* P_{RP27} -*TRI10* transformant (*TRI10*-OE/*tri6*). **e** The expression of *TRI6* and *TRI5* assayed by qRT-PCR with RNA isolated from 3-day-old LTB cultures of the wild-type strain PH-1, P_{RP27} -*TRI6* transformant (*TRI6*-OE), *tri10* mutant, and *tri10* P_{RP27} -*TRI6* transformant (*TRI6*-OE/*tri10*). **f** DON production in 7-day-old LTB cultures of the labelled strains. **g** LTB cultures of the labelled strains were examined for bulbous structures (marked with arrows). Bar =20 μ m. **h** The expression of anti*TRI5* assayed by qRT-PCR with RNA isolated from 3-day-old LTB cultures of *tri6* mutant (arbitrarily set to 1), and *tri6* P_{RP27} -*TRI10* transformant (*TRI10*-OE/*tri6*). Mean and standard deviation were estimated with data from three independent replicates ($n = 3$). For DON production, different letters indicate significant differences based on the one-way ANOVA followed by Turkey's multiple range test ($P < 0.05$).

Line 187: no evidence is provided to say “degradation of sense transcripts of *TRI5*....”

Response: This sentence was revised to “These results indicate that overexpression of antisense-*TRI5* reduced the sense transcripts of *TRI5* and is inhibitory to DON biosynthesis”.

Because of abundant sense transcripts in the region without antisense transcripts but only rare sense transcripts in the region pairing with the antisense transcripts (Fig. 1d), we assumed that the pairing of antisense-*TRI5* with sense-*TRI5* transcripts will result in the degradation of dsRNA as usual. However, we agree that we have no direct experimental data on the degradation of sense transcripts of *TRI5*.

2. The RNAseq data also suggested that lcrRNAs are involved in the regulation of *TRI5*.

Q: Did comparison to other RNAseq database reveal potential CDS/small ORFs in the intergenic region between *TRI6* and *TRI5*?

Response: Thanks for the suggestion. There are no small ORFs or potential CDSs in the intergenic region between *TRI6* and *TRI5* in the RNA-seq data that have been published.

- Please indicate the version of PH1 reference genome used.

Response: Revised as suggested.

- Figure 1: please indicate transcript sizes and the distance between the genes in the cluster (specifically *TRI5* and 6).

Response: Fig. 1 was revised to show the sizes of sense and antisense transcripts of *TRI5* and *TRI6* transcripts (Fig. 1d) and the distance among *TRI* genes in the cluster (Fig. 1a, c). A scale bar of 1 kb was added.

Q: How was TSS determined for the RNA5P in Figure 4.

Response: The transcriptional start site (TSS) for RNA5P is based on the first nucleotide of RNA5P transcripts in the RNA-seq data.

- provide the sequence of the lcrRNA (start and end) and how did they decide which frame to use for tagging the RNA5P?

Response: Related information was added in Supplementary Fig. 5 as suggested. The ORF was predicted with the ExpASy translate tool (web.expasy.org/translate/). The AS (alternative splicing) does not affect the stop codon position. Therefore, the epitope tag was inserted at base 617 (before the stop codon TAA).

- To suggest that “RNA5P does not encode proteins” (line 208) is premature (taken previous point into account). The authors could also use in vitro translation.

Response: This sentence was revised to “These results suggest that RNA5P does not encode a functional proteins and likely act as a lncRNA in *F. graminearum*”. In addition to the failure to detect the RNA5P-3xFLAG fusion protein on western blots with the anti-FLAG antibody, we generated an RNA5P^M transformant in which an adenine (A) was inserted after the base 503 of RNA5P during the revision of this manuscript. If RNA5P encodes a small protein, the functions of this protein will be disrupted by the insertion event that causes a frame-shift mutation. The resulting RNA5P^M transformant was normal in *TRI5* expression and DON production, which provides another line of evidence that RNA5P may not encode a functional protein. Related data was presented in revised Fig. 4.

The authors replaced the *TRI5* promoter with *TRI12* promoter

Q: did RNA seq data reveal any potential transcripts from the *TRI12* promoter region in the WT strain?

Response: Transcripts from the *TRI12* promoter region were not detected based on the RNA-seq.

In situ overexpression by *TrpC* promoter.

Please ensure the transformants TR1, TR2, ATR1, ATR2 in Figure 7 is referenced in the text.

Response: Revised as suggested. All these strains were properly described in the text.

Discussion:

Could use more editing and focus. Please rephrase:

Line 283: ‘global regulator’ is not defined by the quantity of genes, but by the number of pathways it influences.

Response: This sentence was revised into “*TRI10* is considered as a pathway-specific regulator while *TRI6* functions as a regulator to regulate other genes functionally related to plant infection”.

Line 289: both *TRI6* and *TRI10* are necessary for DON regulation. Its presumptuous to say one is more important than the other.

Response: This sentence was revised to: ‘It is possible that *TRI6* regulates more genes related to plant infection but *TRI10* regulates many more genes than *TRI6* in DON-producing, axenic cultures.’

Line 297: clarify “qRT-PCR assays that do not distinguish sense and antisense.....”

Response: This sentence was revised to: “Sense transcripts are not distinguished from antisense transcripts by conventional microarray analysis and qRT-PCR assays”. Conventional microarray analysis and RT-PCR are not sense-specific and detect both sense and antisense transcripts.

Line 308- check Shostak et al., <https://doi.org/10.1111/mmi.14575>

Response: The suggested reference was cited and related description was added in the revised manuscript.

Line 316: no evidence for this

Response: This sentence was revised.

Line 326: use more updated references, specifically the consensus statement regarding nomenclature and function of lcrRNAs <https://doi.org/10.1038/s41580-022-00566-8>. doi: 10.3389/fmicb.2021.638617

Response: References were carefully checked and updated.

Line 337: DON is made by specific species of fungi and the statement in line 337 is a “truism”

Response: This sentence was revised. To our knowledge, RNA5P is the first lncRNA known to regulate secondary mechanisms in filamentous fungi.

Line 338: secondary mechanism or secondary metabolism?

Response: Revised to secondary metabolism.

Line 338-339: rephrase. Consider “In yeasts, plants, and animals cis-acting some lncRNAs mediate.....(PRC2), while others repress.....”

Response: Thanks. This sentence was revised to “In yeast, plants, and animals, some cis-acting lncRNAs mediate gene silencing through the recruitment of polycomb repressive complex 2 (PRC2), while others repress the transcription of target genes by nucleosome rearrangements or formation of RNA-DNA hybrids”.

Other issues:

line 148: “particularly the full-length TRI10 transcripts (Fig 1c)” – NOT shown

Response: Corrected.

Line 168: typo: coving to covering

Response: Corrected.

Figure 4c: what is L3F?

Response: Strain L3F is PRP27-RNA5P-3×FLAG transformant. We have added this information in the revised figure legend.

Figure 4: legend. What is FgESA1-3 x FLAG?

Response: *FgESA1* was used as a control for western blot analysis. A detailed description of *FgESA1* was added in the figure legend. The following sentence was added to Results. “As the control, the expression of FgEsa1 histone acetyltransferase was detected in the *FgESA1*-3xFLAG transformant (Jiang et al., *PLoS Genet.*, 2020) under the same conditions (Fig. 4d)”.

Figure 6a. primer pair combination on top of the gel should read P27F/5PR

Response: Thanks. As suggested by reviewer 1, this image was deleted in revised Fig. 6.

Reviewer #1 (Remarks to the Author):

The manuscript was adequately revised in response to reviewers. I recommend publication of this manuscript.

Reviewer #3 (Remarks to the Author):

the revised document satisfies all the queries i had posed. i recommend publication upon few corrections (see below).

i have recommended few suggestions, but i strongly recommend rephrasing/editing some of other sentences in the Discussion section

Line 317: "have observed in mutants" to have been observed in mutants

Line 319-320: In *F. sporotrichioides*, TRI10 acts as the master regulator and it regulates the expression of TRI6. Where TRI10 regulates many other genes, TRI6 appears to be more specific for regulating TRI involved in T-2 toxin

Grammatically incorrect and requires rephrasing.

Line 322: deletion of TRI10 appeared to have no obvious effect on TRI6 expression by Seong and colleagues. Consider "deletion of TRI10 seemingly displayed no obvious effect on TRI6 expression as shown by Seong and colleagues".

Line 325: consider: "In comparison with the wild type, antisense transcripts of TRI6 in the tri10 mutant became more abundant as detected by microarray analysis and qRT-PCR assays".

Line 329: Therefore, TRI6 and TRI10 may regulate each other's expression in LTB cultures with TRI10 as the master, global regulator in *F. graminearum*

Line 332: change Tri6 to TRI6 for consistency

Line 332: consider replacing "Furthermore" with "in concordance"

Line 334: replace, with ;

Line 358: consider: "Overall, the regulation on TRI genes is multilayered and involves transcription factors, antisense transcripts, and chromosomal organization."

Below are our point-by-point responses to reviewers' comments.

Reviewer #1 (Remarks to the Author):

The manuscript was adequately revised in response to reviewers. I recommend publication of this manuscript.

Response: Thanks for the comments.

Reviewer #3 (Remarks to the Author):

the revised document satisfies all the queries i had posed. i recommend publication upon few corrections (see below).

i have recommended few suggestions, but i strongly recommend rephrasing/editing some of other sentences in the Discussion section.

Response: Thanks for the comments and suggestions.

Line 317: "have observed in mutants" to have been observed in mutants

Response: Revised as suggested.

Line 319-320: In *F. sporotrichioides*, TRI10 acts as the master regulator and it regulates the expression of TRI6. Where TRI10 regulates many other genes, TRI6 appears to be more specific for regulating TRI involved in T-2 toxin
Grammatically incorrect and requires rephrasing.

Response: This sentence has been revised into "In *F. sporotrichioides*, *TRI10* functions as the principal regulatory element, orchestrating the expression of *TRI6*. While *TRI10* regulates many other genes, *TRI6* exhibits a more specific role in governing the *TRI* genes associated with T-2 toxin production".

Line 322: deletion of TRI10 appeared to have no obvious effect on TRI6 expression by Seong and colleagues. Consider "deletion of TRI10 seemingly displayed no obvious effect on TRI6 expression as shown by Seong and colleagues".

Response: Revised as suggested.

Line 325: consider: "In comparison with the wild type, antisense transcripts of TRI6 in the tri10 mutant became more abundant as detected by microarray analysis and qRT-PCR assays".

Response: Revised as suggested.

Line 329: Therefore, TRI6 and TRI10 may regulate each other's expression in LTB cultures with TRI10 as the master, global regulator in *F. graminearum*

Response: Revised as suggested.

Line 332: change Tri6 to TRI6 for consistency

Response: Revised as suggested.

Line 332: consider replacing “Furthermore” with “in concordance”

Response: Revised as suggested.

Line 334: replace, with ;

Response: Revised as suggested.

Line 358: consider: “Overall, the regulation on TRI genes is multilayered and involves transcription factors, antisense transcripts, and chromosomal organization.”

Response: Revised as suggested.